# Neuronal octopamine signaling regulates mating-induced germline stem cell increase in female *Drosophila melanogaster*

Yuto Yoshinari[1], Tomotsune Ameku[1†], Shu Kondo[2], Hiromu Tanimoto[3], Takayuki Kuraishi[4,5], Yuko Shimada-Niwa[6], Ryusuke Niwa[6,7]*

[1]Graduate School of Life and Environmental Sciences, University of Tsukuba, Tsukuba, Japan; [2]Invertebrate Genetics Laboratory, National Institute of Genetics, Mishima, Japan; [3]Graduate School of Life Sciences, Tohoku University, Sendai, Japan; [4]Faculty of Pharmacy, Institute of Medical, Pharmaceutical and Health Sciences, Kanazawa University, Kanazawa, Japan; [5]AMED-PRIME, Japan Agency for Medical Research and Development, Tokyo, Japan; [6]Life Science Center for Survival Dynamics, Tsukuba Advanced Research Alliance (TARA), University of Tsukuba, Tsukuba, Japan; [7]AMED-CREST, Japan Agency for Medical Research and Development, Tokyo, Japan

*For correspondence: ryusuke-niwa@tara.tsukuba.ac.jp

Present address: [†]MRC Clinical Sciences Centre, Imperial College London, Hammersmith Campus, London, United Kingdom

Competing interests: The authors declare that no competing interests exist.

**Abstract** Stem cells fuel the development and maintenance of tissues. Many studies have addressed how local signals from neighboring niche cells regulate stem cell identity and their proliferative potential. However, the regulation of stem cells by tissue-extrinsic signals in response to environmental cues remains poorly understood. Here we report that efferent octopaminergic neurons projecting to the ovary are essential for germline stem cell (GSC) increase in response to mating in female *Drosophila*. The neuronal activity of the octopaminergic neurons is required for mating-induced GSC increase as they relay the mating signal from sex peptide receptor-positive cholinergic neurons. Octopamine and its receptor Oamb are also required for mating-induced GSC increase via intracellular $Ca^{2+}$ signaling. Moreover, we identified Matrix metalloproteinase-2 as a downstream component of the octopamine-$Ca^{2+}$ signaling to induce GSC increase. Our study provides a mechanism describing how neuronal system couples stem cell behavior to environmental cues through stem cell niche signaling.

## Introduction

Animal tissues are built from cells originally derived from stem cells (*Spradling et al., 2001*). During normal development and physiology, this robust stem cell system is precisely regulated (*Drummond-Barbosa, 2008*). Conversely, the dysregulation of these cells can result in abnormal tissue integrity and lead to deleterious diseases. Previous studies have revealed that many types of stem cells reside in a specialized microenvironment, or niche, where they are exposed to local signals required for their function and identity (*Morrison and Spradling, 2008*; *Spradling et al., 2001*). Recently, researchers have demonstrated how stem cell activity is regulated by tissue-extrinsic signals, such as hormones and neurotransmitters. For instance, in mammals, hematopoietic stem cells, mammary stem cells, muscle stem cells, and neural stem cells are influenced by sex hormones such as estrogen (*Asselin-Labat et al., 2010*; *Bramble et al., 2019*; *Kim et al., 2016*; *Nakada et al., 2014*). Retinoic acid and thyroid hormone play essential roles in the differentiation of testicular stem cells and neural stem cells, respectively (*Gothié et al., 2017*; *Ikami et al., 2015*). In addition,

**eLife digest** Stem cells have the unique ability to mature into the various, specialized groups of cells required for organisms to work properly. Local signals released by the tissues immediately surrounding stem cells usually trigger this specialization process. However, recent studies have revealed that external signals, such as hormones or neurotransmitters (the chemicals used by nerve cells to communicate), can also control the fate of stem cells. This is particularly the case during development, or in response to events such as injury.

In the right conditions, germline stem cells can specialize into the egg or sperm required for many animals to reproduce. In fruit flies for example, the semen contains proteins that activate a cascade of molecular events in the female nervous system, ultimately resulting in female germline stem cells multiplying in the ovaries after mating. Yet, exactly how this process takes place was still unclear. To investigate this question, Yoshinari et al. focused on nerve cells in the fruit fly ovary which produce a neurotransmitter called octopamine.

The experiments assessed changes in the ovaries of female fruit flies after mating, piecing together the sequence of events that activate germline stem cells. This showed that first, mating triggers the release of octopamine from the nerve cells. In turn, this activates a protein called Oamb, which is studded through the membrane of cells present around germline stem cells. Turning on Oamb prompts a cascade of molecular events which include an enzyme called Matrix metalloproteinase 2 regulating the signal sent from the local environment to germline stem cells.

As mammals use a neurotransmitter similar to octopamine, future fruit fly studies could shed light on how neurotransmitters activate stem cells in other animals. Ultimately, unravelling the way external signals trigger the specialization process may offer insight into how diseases arise from uncontrolled stem cell activity.

mesenchymal stem cell proliferation is stimulated by adrenaline (*Wu et al., 2014*). However, the when, how, and why these humoral factors are produced, circulated, and received during stem cell regulation remain to be elucidated.

The ovaries of the fruit fly *Drosophila melanogaster* are an excellent model system on how stem cell lineages are shaped by both local niche signals and tissue-extrinsic signals (*Drummond-Barbosa, 2019*). *D. melanogaster* ovary is composed of 16–20 chains of developing egg chambers called ovarioles. The anterior-most region of which, known as the germarium, contains germline stem cells (GSCs) that give rise to the eggs (*Figure 1A and B*). GSCs are adjacent to the somatic niche cells, which comprises cap cells, escort cells, and terminal filament cells (*Figure 1A*). After GSC divides, one daughter cell that remains attached to the niche cells retains its GSC identity, whereas the remaining daughter cells are displaced away from the niche cells and differentiate into cystoblast (CB). Each CB then undergoes differentiation into 15 nurse cells and one oocyte in each egg chamber, which is surrounded by somatic follicle cells.

GSC niche produces and secretes several local niche signals that regulate the balance between GSC self-renewal and differentiation (*Hayashi et al., 2020*; *Kirilly and Xie, 2007*; *Spradling et al., 2011*). For example, bone morphogenetic protein (BMP) ligands Decapentaplegic (Dpp) and Glass bottom boat (Gbb) are produced from the niche cells and directly activate BMP receptors in GSCs, leading to the repression of the differentiation inducer, *bag-of-marbles* (*bam*) (*Morrison and Spradling, 2008*; *Zhang and Cai, 2020*). Recent *D. melanogaster* GSC studies have also contributed to understanding of the systemic regulation of stem cell proliferation and maintenance in response to external environmental cues (*Ables and Drummond-Barbosa, 2017*; *Drummond-Barbosa, 2019*; *Lin and Hsu, 2020*; *Yoshinari et al., 2019*). For example, protein restriction results in a reduction in GSC division, which is mediated by *Drosophila* insulin-like peptides (DILPs) (*LaFever, 2005*). In addition, nutrients influence GSC maintenance via the adipocyte metabolic pathway (*Armstrong and Drummond-Barbosa, 2018*; *Matsuoka et al., 2017*).

Besides nutrients, we have recently found out that mating is another external cue that significantly affects *D. melanogaster* GSC increase. Mated females show a dramatic increase in egg production, as well as GSC, which is induced by a male-derived peptide from the seminal fluid called sex peptide (SP) (*Kubli, 2003*; *Yoshinari et al., 2019*). SP is received by its specific receptor, sex peptide

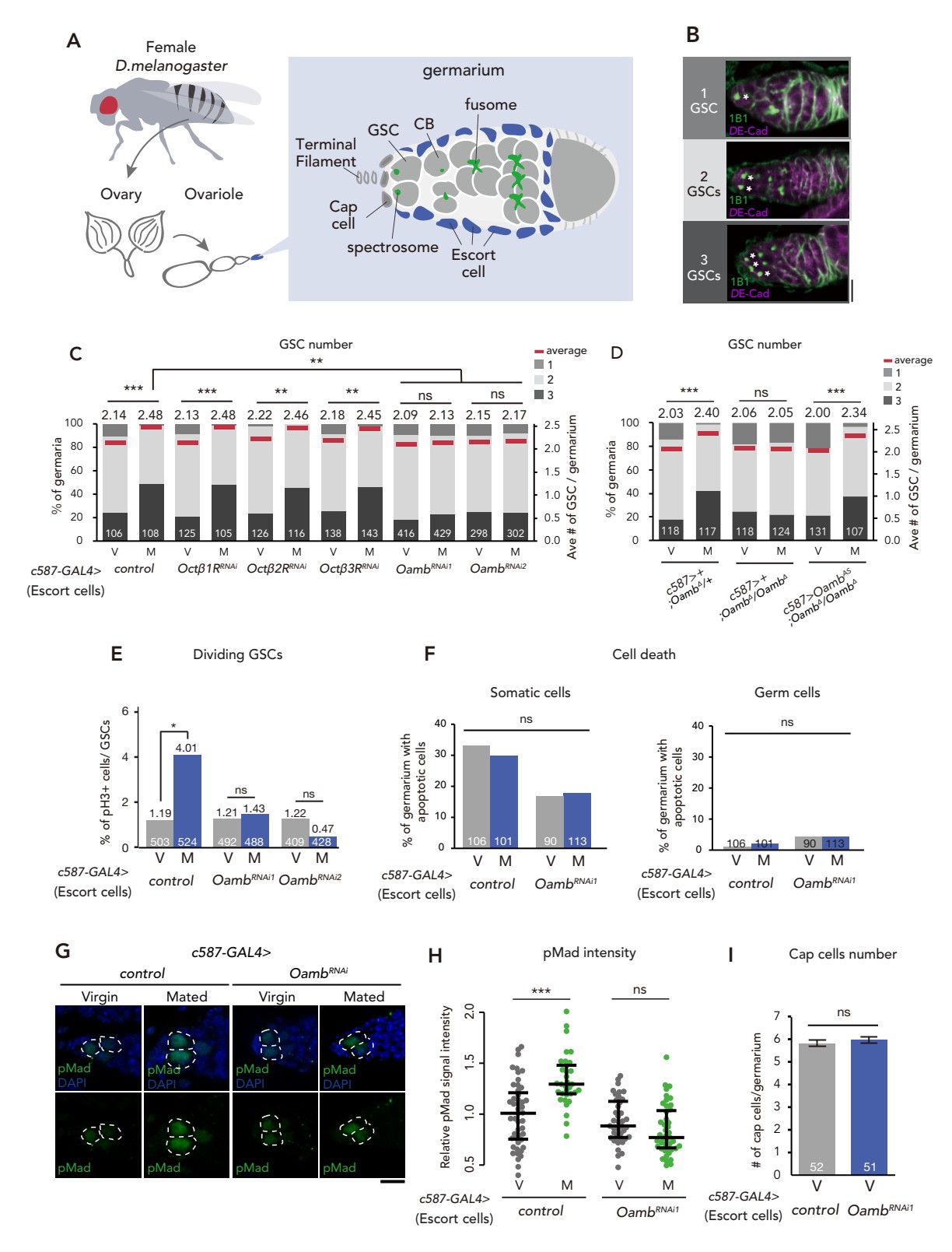

**Figure 1.** Post-mating GSC increase requires Oamb in the escort cells. (**A**) A schematic representation of *Drosophila* germarium. GSCs reside in a niche consisting of somatic cells such as cap cells, terminal filament cells, and escort cells and are identifiable by their stereotypical spectrosome morphology and location (adjacent to cap cells). GSC division produces one self-renewing daughter and one cystoblast (CB) that differentiates into a germline cyst. (**B**) Representative images of wild-type ($w^{1118}$) female adult germariums, containing 1, 2 and 3 GSCs from top to bottom. The samples were stained with

*Figure 1 continued on next page*

*Figure 1 continued*

monoclonal antibody 1B1 (green) and anti-*D*E-cadherin (magenta), which stain the spectrosome and overall cell membranes, respectively. GSCs are indicated by asterisk. Scale bar, 20 µm. (C–D) Frequencies of germaria containing 1, 2, and 3 GSCs (left vertical axis) and the average number of GSCs per germarium (right vertical axis) in virgin (V) and mated (M) female flies. *c587>+* flies were used as the control in D. (E) The ratio of pH3$^+$ GSCs per total GSCs. (F) The ratio of apoptotic (Dcp-1$^+$) somatic cells and germ cells per germarium. *c587>+* flies were used as the control. (G) Representative images of adult female germaria immunostained with anti-pMad antibody (green) and DAPI (blue) are shown. GSCs are outlined with dotted lines. Scale bar, 10 µm. (H) Quantification of relative pMad intensity levels in the GSCs (i.e. virgin (V), mated (M)) as normalized to the pMad intensity in CBs. Each sample number was at least 25. The three horizontal lines for each sample indicate lower, median, and upper quartiles. (I) The number of cap cells per germarium in the control and *Oamb* RNAi driven by *c587-GAL4*. Values on the y-axis are presented as the mean with standard error of the mean. *c587>+* flies were used as the control. For C-F, and I the number of germaria analyzed is indicated inside the bars. Wilcoxon rank sum test with Holm's correction was used for C, D, H, and I. Fisher's exact test with Holm's correction was used for E and F. ***p≤0.001, **p≤0.01, and *p≤0.05; NS, nonsignificant (p>0.05). All source data are available in *Source data 1* and *2*.

The online version of this article includes the following figure supplement(s) for figure 1:

**Figure supplement 1.** Oamb acts in the escort cells for post-mating GSC increase.
**Figure supplement 2.** Expression of *Oamb knock-in GAL4.*
**Figure supplement 3.** Oamb in the escort cells is necessary on mating-induced BMP signaling increase.

receptor (SPR), in a small subset of SPR-positive sensory neurons (SPSNs), which are located in the uterine lumen and send afferent axons into the tip of the abdominal ganglion (*Häsemeyer et al., 2009*; *Yapici et al., 2008*). The SP-SPR signaling in the SPSNs stimulates the biosynthesis of the ovarian insect steroid hormones (ecdysteroids), which play an essential role in mating-induced GSC increase (*Ameku et al., 2017*; *Ameku and Niwa, 2016*; *Uryu et al., 2015*). Because SPSNs do not directly innervate into the ovary, it is hypothesized that a signal and its signaling pathway are involved in bridging the gap between SPSNs and GSCs. However, it is still unclear how mating information is transmitted from SPSNs to GSCs at the molecular and cellular levels.

Here, we present a series of new findings that reveal a novel and fundamental neuronal mechanism connecting SPSNs and GSCs to regulate mating-induced GSC increase. We demonstrate that a small subset of neurons directly innervating into the ovary plays an indispensable role in regulating mating-induced GSC increase. These neurons produce the monoamine neurotransmitter, octopamine (OA), the insect equivalent of noradrenaline (*Roeder, 2005*). We also show that the neuronal activity of the OA-producing neurons is required for mating-induced GSC increase. Moreover, we find that the OA directory activates GSC increase through its receptor, octopamine receptor in mushroom body (Oamb), followed by Ca$^{2+}$ signaling in the ovarian escort cells. Furthermore, OA/Oamb signaling requires Matrix metalloproteinase 2 (Mmp2) to activate GSC increase in the ovarian escort cells. Finally, we show that SPSNs relay the mating signal to the ovary-projecting OA neurons via nicotinic acetylcholine receptor signaling. Taken together, we propose a novel efferent neuronal pathway that transmits mating stimulus to the GSC to control stem cell number. Our study provides a mechanism describing how neuronal system couples stem cell behavior to environmental cues, such as mating, through stem cell niche signaling.

## Results

### Mating-induced GSC increase requires the octopamine receptor Oamb in ovarian escort cells

As a candidate signal that bridges between SPSNs and GSC increase, we focused on the biogenic amine, OA, because a part of the octopaminergic neurons innervate to the ovary and the oviduct (*Heifetz et al., 2014*; *Rezával et al., 2014*). Moreover, it has been reported that OA and Oamb signaling regulate ovulation process and ovarian-muscle contraction (*Deady and Sun, 2015*; *Monastirioti, 2003*; *Rezával et al., 2014*). We first conducted transgenic RNAi screen against 4 OA receptor genes with *c587-GAL4*, which is active in the ovarian-somatic cells, including the escort cells of the germarium (*Manseau et al., 1997*). In control females, the mated ones exhibited an increase in GSC number as we have reported previously (*Ameku and Niwa, 2016*; *Figure 1C*). In contrast, *c587-GAL4*–mediated *Oamb* knock-down (*c587 >Oamb^RNAi*) showed significantly impaired mating-induced GSC increase (*Figure 1C*). This phenotype was observed with two independent *UAS-Oamb-RNAi* strains (*Oamb^RNAi1* and *Oamb^RNAi2*) (*Figure 1C*), each of which targeted a different region in

the *Oamb* mRNA. The specificity of *Oamb* was also confirmed by the fact that the *c587-GAL4*–driven transgenic RNAi of other octopamine receptor genes (*Octβ1R*, *Octβ2R* and *Octβ3R*) (*Ohhara et al., 2012*) had no significant effect on the GSC number between virgin and mated females (*Figure 1C*). Therefore, Oamb has a pivotal role in mating-induced GSC increase.

We next examined in which ovarian-somatic cells Oamb regulates mating-induced GSC increase with several *GAL4* lines that are active in the specific ovarian-somatic cells. Previous studies have demonstrated that Oamb in mature follicle cells and in the oviduct has a significant role in ovulation (*Deady and Sun, 2015*; *Lee et al., 2009*; *Lee et al., 2003*). Therefore, it is possible that *Oamb* may indirectly induce GSC increase via Oamb-mediated ovulation processes. However, mating-induced GSC increase was not impaired by *Oamb* RNAi in the stage-14 follicle cells by *R44E10-GAL4* (*Deady and Sun, 2015*) (*R44E10 >Oamb^{RNAi}*), in the stage 9–10 follicle cells by *c355-GAL4*, *c306-GAL4*, and *slbo-GAL4* (*Barth et al., 2012*), or in the common oviduct by *RS-GAL4* (*Lee et al., 2003*) (*RS >Oamb^{RNAi}*) (*Figure 1—figure supplement 1A,B and E*). These data suggest that mating-induced GSC increase is independent from the ovulation process.

Consistent with the observation using *c587-GAL4*, *Oamb* RNAi by *Traffic jam (Tj)-GAL4* (*Olivieri et al., 2010*) (*Tj >Oamb^{RNAi}*), *R13C06-GAL4,* and *109–30-GAL4* (*Sahai-Hernandez and Nystul, 2013*), which are active in the pan-ovarian-somatic cells, the escort cells, and the germarium follicle cells, respectively, also resulted in the failure of mating-induced GSC increase (*Figure 1—figure supplement 1C and E*). On the other hand, *Oamb* RNAi in the cap cells (*bab >Oamb^{RNAi}*) and germ cells (*nos >Oamb^{RNAi}*) had no effect on GSC increase (*Figure 1—figure supplement 1D*). These results suggest that Oamb in the escort cells or the follicle cells of the germarium plays an essential role in mating-induced GSC increase.

It must be noted that *c587-GAL4* and *Tj-GAL4* are expressed not only in the ovarian-somatic cells but also in the nervous system (*Ameku et al., 2018*). Moreover, *Oamb* is expressed in the nervous system (*Han et al., 1998*). However, *Oamb* RNAi in the nervous system (*nSyb >Oamb^{RNAi}*) did not affect the mating-induced GSC increase (*Figure 1—figure supplement 1D*), suggesting that the impairment of GSC increase of *c587 >Oamb^{RNAi}* or *tj >Oamb^{RNAi}* is not due to gene knock-down in neuronal cells but rather in the ovarian-somatic cells. These data also support our idea that Oamb in the ovarian-somatic cells regulate mating-induced GSC increase.

To confirm the role of *Oamb* in mating-induced GSC increase, we generated a *Oamb* complete loss-of-function genetic allele by Clustered Regularly Interspaced Short Palindromic Repeats (CRISPR)/CRISPR-associated protein 9 (Cas9) technology (*Kondo and Ueda, 2013*; *Figure 1—figure supplement 1F*). Similar to *Oamb* RNAi females, *Oamb* homozygous mutant females (*c587>+; Oamb^Δ/Oamb^Δ*) did not exhibit mating-induced GSC increase (*Figure 1D*). In addition, the GSC increase of *Oamb^Δ/Oamb^Δ* was restored by overexpression of *Oamb* in the escort cells (*c587 >Oamb^{AS}; Oamb^Δ/Oamb^Δ*). These findings are all consistent with the idea that Oamb in escort cells modulates GSC increase after mating.

We also examined *Oamb* expression in the ovarian-somatic cells by two *Oamb*-knock-in *GAL4* lines (*Deng et al., 2019*; *Kondo et al., 2020*). However, we could not detect any reliable signals in the germarium or the mature follicle cells via these *GAL4* lines with *UAS-GFP* and *UAS-Stinger* lines (*Figure 1—figure supplement 2A,B,C and D*). We speculate that this may be due to lower amounts of *Oamb* transcript in the germarium.

## Oamb in escort cells is required for GSC increase after mating

Because mating-induced GSC increase is accompanied by GSC division (*Ameku and Niwa, 2016*), We next examined whether Oamb in the escort cells is involved in GSC division after mating. We determined the number of GSCs during the M phase by staining using anti-phospho-Histone H3 (pH3) in control and *c587 >Oamb^{RNAi}* females. In control female flies, mating increased the frequency of GSCs in the M phase (*Figure 1E*), whereas in *c587 >Oamb^{RNAi}* flies, this was not observed. We also monitored the fraction of apoptotic cells in the germarium by staining with anti-cleaved death caspase-1 (Dcp-1), a marker for apoptotic cells (*Song et al., 1997*). The number of apoptotic cells in the germarium did not change in *c587 >Oamb^{RNAi}* female flies compared with controls (*Figure 1F*), suggesting that Oamb activates GSC increase by pushing the cell cycle of GSCs and that the lack of mating-induced GSC increase in *Oamb* RNAi is not due to the enhancement of cell death.

Our previous studies revealed that mating-induced GSC increase is mediated by GSC niche signals (*Ameku et al., 2018*; *Ameku and Niwa, 2016*). In particular, Decapentaplegic (Dpp), the fly counterpart to bone morphogenetic protein (BMP), is the essential niche signal (*Spradling et al., 2011*; *Xie and Spradling, 1998*). We therefore examined whether *Oamb* knock-down affects Dpp signaling in GSCs by measuring the level of phosphorylated Mad (pMad), a readout of Dpp signaling activation (*Chen and McKearin, 2003*; *Raftery and Sutherland, 1999*). We confirmed that mating induced the increase in pMad level in GSCs, whereas mating did not increase pMad levels in *c587 >Oamb*$^{RNAi}$ animals (*Figure 1G and H*). We also confirmed that mating increased the signal intensity of *Daughters against dpp (Dad)-LacZ*, a reporter gene that reflect an expression of the BMP target gene *Dad*, while *Oamb* knock-down in the escort cells impaired the increase of *Dad-LacZ* signal after mating (*Figure 1—figure supplement 3A and B*). These results suggest that mating activates the BMP signal in GSCs through Oamb in the escort cells, thereby resulting in the increase in GSCs.

Further, we determined the number of cap cells, which are critical components of the GSC niche (*Xie and Spradling, 1998*). *c587 >Oamb*$^{RNAi}$ did not change the number of cap cells in virgin nor mated female flies, suggesting that *Oamb* knock-down does not affect the overall architecture of the niche (*Figure 1I*). Overall, Oamb in the escort cells plays a pivotal role in mating-induced GSC increase.

## OA administration is sufficient to induce GSC increase and BMP signaling in GSCs in an Oamb-dependent manner

To examine whether OA is received in the ovary but not in other organs to induce GSC increase, we cultured dissected virgin ovaries ex vivo with or without purified OA in the culture medium. After incubation for 12 hr, the ovaries cultured with OA had more GSCs as compared to those without OA (*Figure 2—figure supplement 1A*). Whereas the minimal OA concentration to induce ex vivo GSC increase was 1 µM, hereafter we used 100 µM OA because the GSC number plateaued with this concentration (*Figure 2—figure supplement 1A*). This OA-mediated ex vivo GSC increase was not observed in *c587 >Oamb*$^{RNAi}$ virgin ovaries (*Figure 2A*).

We also examined whether OA treatment affects Dpp signaling in GSCs ex vivo by measuring the level of pMad. We confirmed that OA treatment was sufficient to induce the increase in pMad level in GSCs, even in the ex vivo culture system (*Figure 2B*). On the other hand, OA treatment did not increase pMad levels in *c587 >Oamb*$^{RNAi}$ ovaries (*Figure 2B*). These results suggest that OA activates the BMP signal in GSCs through Oamb in the escort cells.

## Ca$^{2+}$ signaling in escort cells is essential for OA-dependent GSC increase

Upon OA binding, Oamb evokes Ca$^{2+}$ release from the endoplasmic reticulum (ER) into the cytosol, leading to a transient increase in intercellular Ca$^{2+}$ concentration ($[Ca^{2+}]_i$) (*Han et al., 1998*). To determine whether OA induces GSC increase via affecting $[Ca^{2+}]_i$ in escort cells, we first monitored $[Ca^{2+}]_i$ using a genetically encoded calcium sensor, GCaMP6s (*Nakai et al., 2001*; *Ohkura et al., 2012*). We dissected the virgin ovaries, in which *GCaMP6s* transgene was expressed driven by *Tj-GAL4* or *c587-GAL4*, cultured them ex vivo, and then observed GCaMP6s fluorescence (*Figure 2C*). We found that 100 µM of OA treatment evoked an increase in GCaMP6s fluorescence in the escort cells and follicle cells, whereas the control medium treatment (0 µM of OA) did not show any increase in fluorescence (*Figure 2D* and *Figure 2—figure supplement 1B*; also see *Videos 1* and *2*). In addition, the OA-mediated increase in GCaMP6s fluorescence intensity was not observed in the germarium of *Oamb* RNAi flies (*Figure 2E*), suggesting that the OA-dependent increase in $[Ca^{2+}]_i$ in the germarium is required for Oamb. Notably, the timeline of signal increase was very slow as this fluorescence increases progressively. This response to OA treatment was similar to a report featuring mature follicle cells of stage-14 oocytes (*Deady and Sun, 2015*).

We also employed another approach utilizing the light-gated cation channel, CsChrimsonn (*Klapoetke et al., 2014*). We prepared *c587-GAL4 >CsChrimson* flies in combination with *nSyb-GAL80* (*Harris et al., 2015*), allowing the expression of *CsChrimson* gene only in the germarium but not in the nervous system. Because CsChrimson requires all trans-retinal (ATR) to form its proper protein conformation (*Wang et al., 2012*), we utilized the flies fed with and without ATR supplement

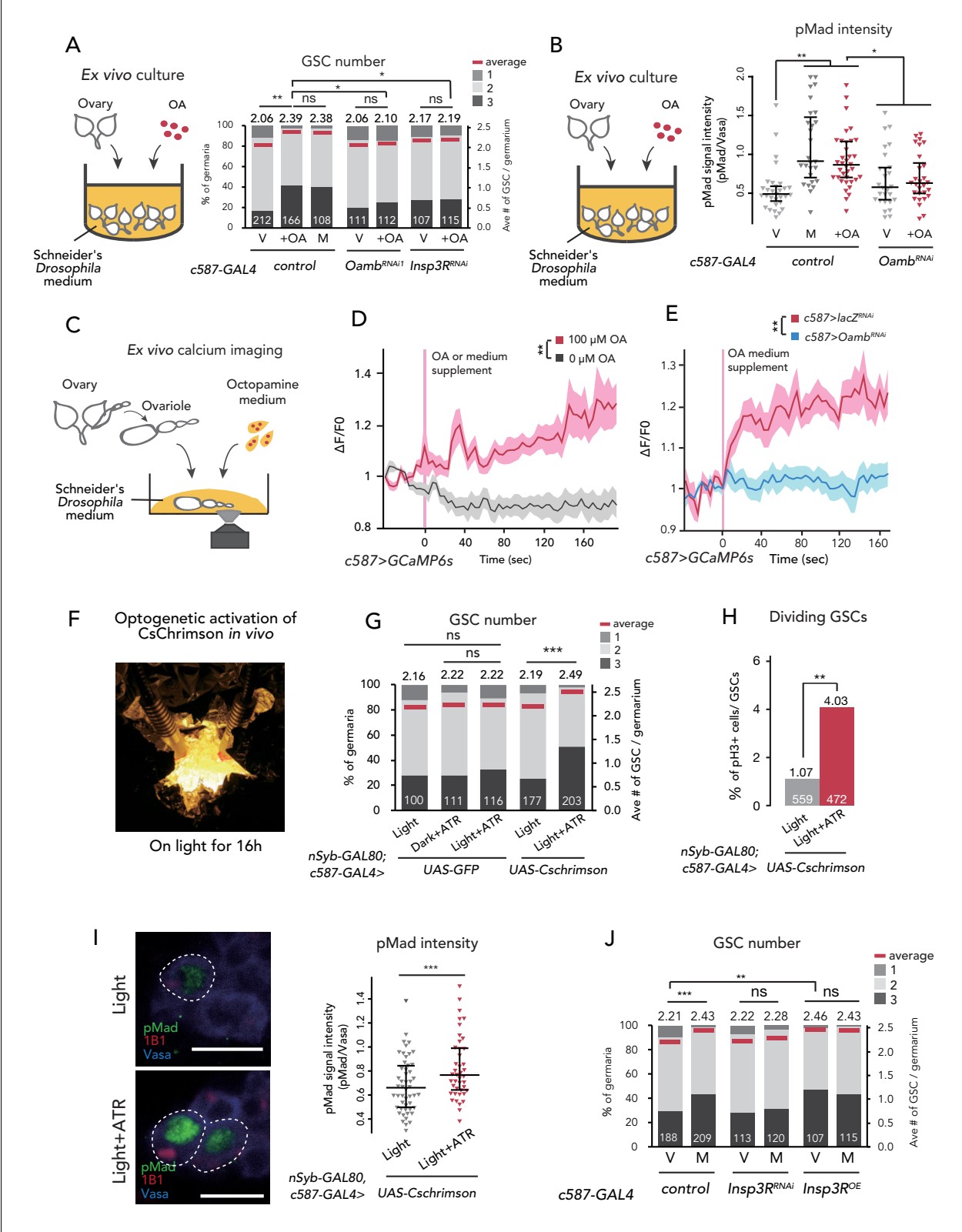

**Figure 2.** $Ca^{2+}$ signaling is necessary for mating-induced GSC increase. (A) Frequencies of germaria containing 1, 2, and 3 GSCs (left vertical axis) and the average number of GSCs per germarium (right vertical axis). The ovaries were dissected from virgin (V), mated (M), and virgin ovaries cultured with OA (+OA). *c587>+* flies were used as the control. The number of germaria analyzed is indicated inside the bars. (B) Quantification of relative pMad intensity levels in the GSCs of ex vivo cultured ovaries (i.e. virgin (V), mated (M), and virgin cultured with OA (+OA)) as normalized to the pMad intensity

*Figure 2 continued on next page*

*Figure 2 continued*

in CBs. For the quantification of pMad intensity, the cell boundaries of GSCs and CBs were determined using anti-Vasa staining. Each sample number was at least 25. The three horizontal lines for each sample indicate lower, median, and upper quartiles. (C) A schematic representation of ex vivo calcium imaging. The dissected ovariole was incubated in Schneider's *Drosophila* medium with or without OA. (D) Changes in the relative fluorescence intensity of GCaMP6s after 200 s without stimulation (n = 8) or with stimulation (n = 10) with 100 µM OA, and (E) with 100 µM OA as control (*c587 >LacZ^{RNAi}*, n = 8) and *c587 >Oamb^{RNAi}* (n = 8) female ovaries. Note that OA significantly increased the calcium response in escort cells, but *Oamb^{RNAi}* impaired the calcium response. Statistical analysis was done at 120 s. (F) Equipment setup for optogenetic activation of ChR. Flies were placed under the light for 16 hr before dissection. (G) Frequencies of germaria containing 1, 2, and 3 GSCs (left vertical axis) and the average number of GSCs per germarium (right vertical axis) with light, with light and all trans-retinal (ATR) or with dark and ATR. Germarium was dissected from virgin females. *nSyb-GAL80; c587 >GFP* flies were used as control. The number of germaria analyzed is indicated inside the bars. (H) The ratio of pH3+ GSCs and total GSCs. The number of GSCs analyzed is indicated inside the bars. (I, left) Representative images of adult female germaria immunostained with anti-pMad antibody (green), anti-1B1 antibody (red), and anti-Vasa antibody (germ cell marker; blue) are shown. GSCs are outlined with dotted lines. (I, right) Quantification of the relative pMad intensity in GSCs, which was normalized to that in CBs. For the quantification of pMad intensity, the cell boundaries of GSCs and CBs were determined using anti-Vasa staining. Each sample number is at least 30. The three horizontal lines for each data sample indicate lower, median, and upper quartiles. (J) Frequencies of germaria containing 1, 2, and 3 GSCs (left vertical axis) and the average number of GSCs per germarium (right vertical axis) in virgin (V) and mated (M) female flies. *c587>+* flies were used as the control. The number of germaria analyzed is indicated inside the bars. Wilcoxon rank sum test with Holm's correction was used for A, B, D, E, G, I, and J. Fisher's exact test was used for H. ***p≤0.001, **p≤0.01, and *p≤0.05; NS, non-significant (p>0.05). All source data are available in *Source data 1*, *2*, and *4*.

The online version of this article includes the following figure supplement(s) for figure 2:

**Figure supplement 1.** OA treatment induces GSC increase.

as experimental and control groups, respectively. When we irradiated an orange-light to *c587-GAL4, nSyb-GAL80 >CsChrimson* flies to induce $Ca^{2+}$ flux in the germarium (*Figure 2F*), GSC increased in the virgin females (*Figure 2G*). In addition, the ratio of GSC in the M phase increased by CsChrimson activation (*Figure 2H*). Moreover, the pMad level in GSCs was increased by CsChrimson activation, suggesting that the forced $Ca^{2+}$ flux is sufficient to induce GSC increase through the upregulation of BMP signaling (*Figure 2I*).

We next confirmed whether the downstream component of $Ca^{2+}$ signaling is involved in the mating-induced GSC increase. The knock-down of *Inositol 3-receptor* (*c587 >Insp3R^{RNAi}*) encoding a protein that releases the stored $Ca^{2+}$ from ER suppressed GSC increase in mated females (*Figure 2J*). Conversely, the overexpression of *Insp3R* in the escort cells increased the GSC number in virgin females (*Figure 2J*). Furthermore, OA-mediated ex vivo GSC increase was not observed in *c587 >Insp3R^{RNAi}* virgin ovaries (*Figure 2A*). Overall, we demonstrated that OA signaling regulates the mating-induced GSC increase by controlling $Ca^{2+}$ signaling in escort cells, which is thereby necessary and sufficient to induce GSC increase.

## Ovarian ecdysteroid signaling is required in the OA-Oamb-$Ca^{2+}$-dependent GSC increase

The biosynthesis and signaling of ecdysteroid in the ovary are required for the mating-induced GSC increase (*Ameku et al., 2017*; *Ameku and Niwa, 2016*). Therefore, we next examined whether ecdysteroid signaling has a pivotal role in the OA-Oamb-$Ca^{2+}$-dependent GSC increase. As we have previously reported, the RNAi of *neverland* (*nvd*), which encodes an ecdysteroidogenic enzyme (*Yoshiyama-Yanagawa et al., 2011*; *Yoshiyama et al., 2006*) in the escort cells, suppressed the mating-induced GSC increase (*Figure 3A*; *Ameku et al., 2017*;

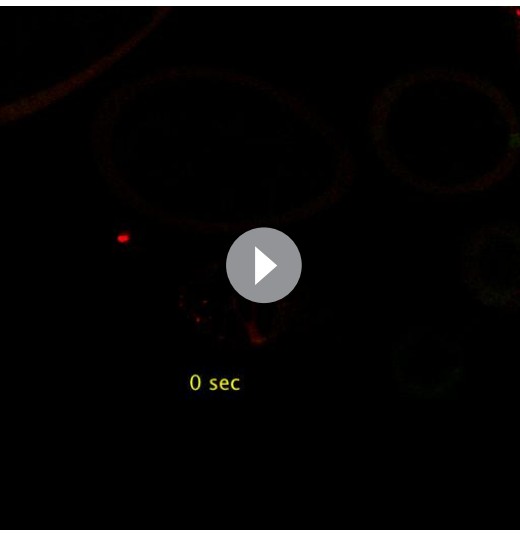

**Video 1.** A video image of the GCaMP6 signal in the ex vivo-cultured germarium without OA administration. A genotype of the germarium was *Tj-GAL4 >UAS-GCaMP6s UAS-mCD8::RFP*.
https://elifesciences.org/articles/57101#video1

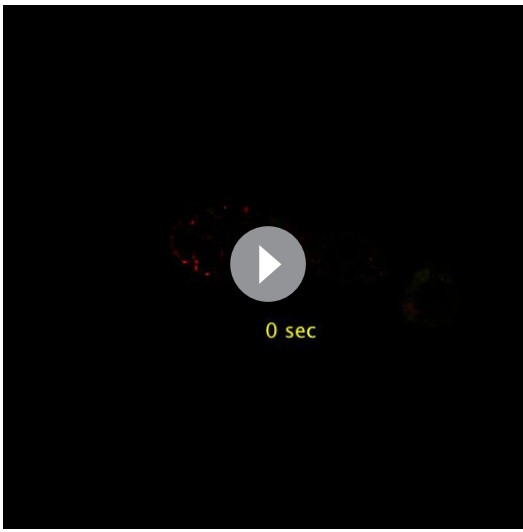

**Video 2.** A video image of the GCaMP6 signal in the ex vivo-cultured germarium with 100 mM OA administration. A genotype of the germarium was *Tj-GAL4 >UAS-GCaMP6s UAS-mCD8::RFP*.
https://elifesciences.org/articles/57101#video2

*Ameku and Niwa, 2016*). We also found a similar phenotype in the RNAi of *ecdysone receptor* (*EcR*) in the escort cells (*Figure 3A*). To assess the requirement of ecdysteroid biosynthesis and signaling in OA-induced GSC increase, we employed an ex vivo experiment. Interestingly, the OA-mediated GSC increase was not observed in *nvd* RNAi ovaries (*c587 >nvd$^{RNAi}$*) (*Figure 3B*). Moreover, this impairment of GSC increase in *c587 >nvd$^{RNAi}$* was restored with the administration of 20-hydroxyecdysone (20E) in the culture media. On the other hand, 20E treatment without OA did not induce GSC increase in the control ovaries (*Figure 3B*). This observation is consistent with our previous study showing that wild-type virgin females fed with 20E did not exhibit any increase in GSCs (*Ameku and Niwa, 2016*). Therefore, 20E is required for the OA-mediated increase in GSCs; however, by itself, 20E is not sufficient to induce GSC increase.

To assess the role of *EcR* in downstream OA signaling, we utilized the temperature-sensitive and null alleles of *EcR* (*EcR$^{A483T}$* and *EcR$^{M54fs}$*, respectively) (*Bender et al., 1997*). At a restrictive temperature of 31°C, the mating-induced GSC increase was suppressed in *EcR$^{A483T}$*/*EcR$^{M54fs}$* flies, consistent with our previous report (*Ameku and Niwa, 2016*; *Figure 3C*). In the ex vivo experiment, the OA-dependent GSC increase was also suppressed in *EcR$^{A483T}$*/*EcR$^{M54fs}$* ovaries (*Figure 3D*). These results suggest that OA-Oamb-Ca$^{2+}$ signaling requires ovarian ecdysteroid signaling.

## Matrix metalloproteinase two acts downstream of OA-Oamb-Ca$^{2+}$ signaling

So far, we have identified four components with indispensable roles in mating-induced GSC increase, namely OA, Oamb, Ca$^{2+}$, and ecdysteroids. Interestingly, recent studies have reported that they are also essential in the follicle rupture in *D. melanogaster* ovary (*Deady and Sun, 2015*; *Knapp and Sun, 2017*). In this process, Matrix metalloproteinase 2 (Mmp2), a membrane-conjugated proteinase, acts downstream of the OA-Oamb-Ca$^{2+}$ signaling pathway in mature follicle cells (*Deady et al., 2015*). Because recent studies have indicated the expression of *Mmp2* in niche cells, including escort cells (*Pearson et al., 2016*; *Wang and Page-McCaw, 2014*), we examined Mmp2 function in the escort cells using RNAi of *Mmp2* with *c587-GAL4*. Similar to *c587 >Oamb$^{RNAi}$*, *c587 >Mmp2$^{RNAi}$* impaired mating-induced GSC increase (*Figure 4A*). Notably, *Mmp2* knock-down in cap cells by *bab-GAL4* also suppressed GSC increase, suggesting that Mmp2 acts in both escort cells and cap cells to induce GSC increase (*Figure 4—figure supplement 1A*). We also found that *Mmp2* RNAi in the ovarian-somatic cells by another *GAL4* (*Tj >Mmp2$^{RNAi1}$*) resulted in the failure of mating-induced GSC increase, whereas *Mmp2* RNAi in the nervous system (*nSyb >Mmp2$^{RNAi1}$*) and in the follicle cells of stage 14 oocytes (*R44E10 >Mmp2$^{RNAi1}$*) had no effect on GSC increase (*Figure 4—figure supplement 1B*). These results suggest that Mmp2 in the escort cells and cap cells are necessary to induce post-mating GSC increase.

In the GSC niche cells in *D. melanogaster*, *Tissue inhibitors of metalloproteinases* (*Timp*) gene encoding an endogenous proteinase inhibitor of Mmp2 is expressed (*Gomis-Rüth et al., 1997*; *Page-McCaw et al., 2003*; *Pearson et al., 2016*). The knock-down of *Timp* in escort cells induced GSC increase even in virgin females, whereas its overexpression suppressed mating-induced GSC increase (*Figure 4B*). Consistent with *Mmp2* knock-down, *Timp* knock-down in cap cells (*bab >TimpRNAi$^{RNAi2}$*) increased the GSC number in virgin females, whereas its knock-down in the follicle cells of stage 14 oocyte (*R44E10 >TimpRNAi$^{RNAi2}$*) had no effect (*Figure 4—figure*

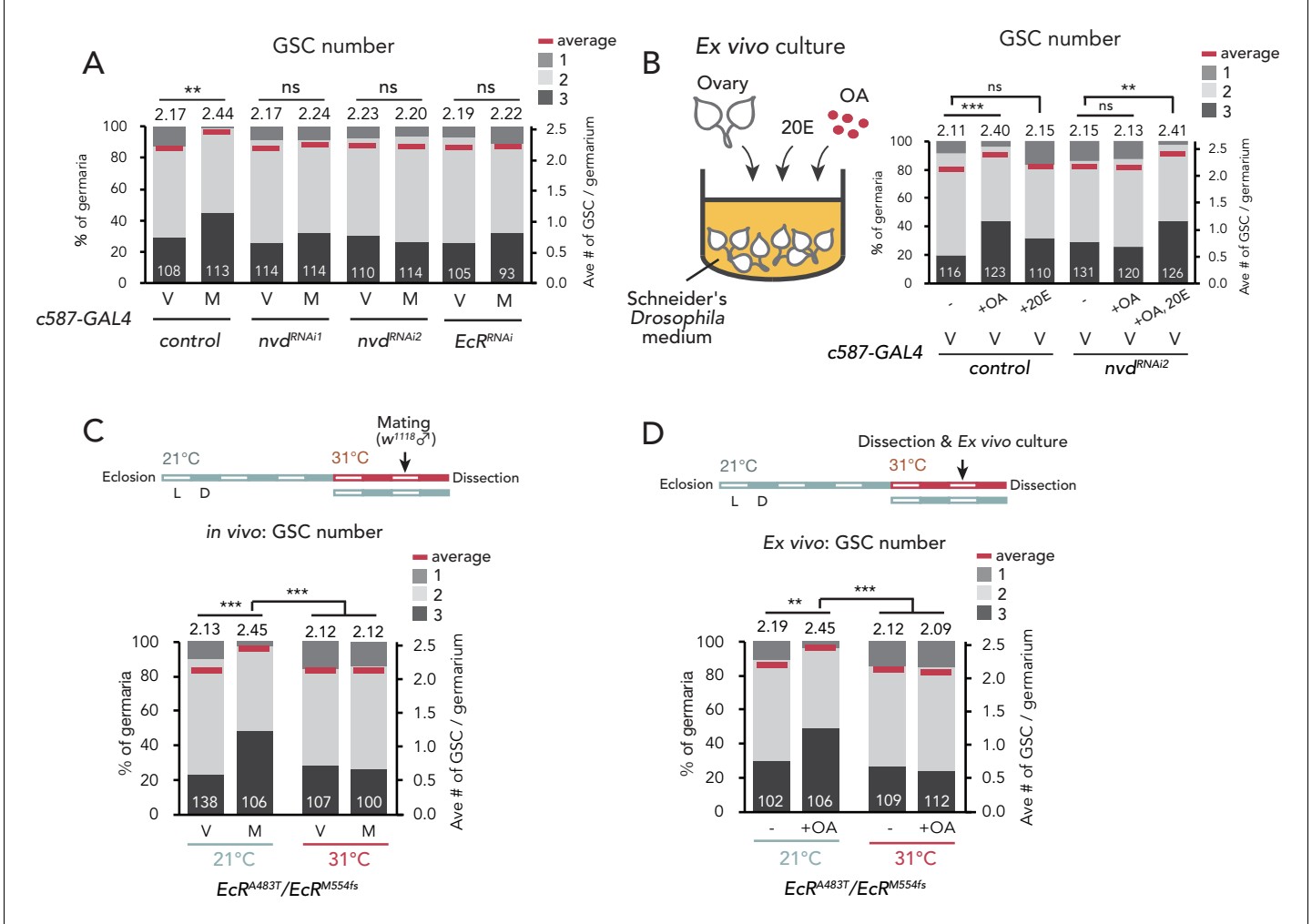

**Figure 3.** Ecdysteroid signaling is necessary for OA-mediated GSC increase. (A–D) Frequencies of germaria containing 1, 2, and 3 GSCs (left vertical axis) and the average number of GSCs per germarium (right vertical axis) in virgin (V) and mated (M) female flies. *c587>+* flies were used as the control. The number of germaria analyzed is indicated inside the bars. (A) GSC number of *nvd* and *EcR* RNAi flies in vivo. (B) Virgin ovaries were cultured ex vivo with or without OA and 20E (+OA, +20E, −), and then the GSC number was determined. (C–D) Experiments using a temperature-sensitive allele *EcR^A483T*. 21°C and 31°C were used as the permissive and restrictive temperatures, respectively. Flies were cultured at 21°C and transferred to 31°C 1 d prior to the assays (L; light, D; dark). (C) GSC number in vivo. (D) Virgin ovaries were cultured ex vivo with or without OA (+OA, −). The number of germaria analyzed is indicated inside the bars. Wilcoxon rank sum test with Holm's correction was used for statistical analysis. ***p≤0.001 and **p≤0.01; NS, non-significant (p>0.05). All source data are available in *Source data 1*.

supplement 1C). These data suggest that Mmp2 activity in the GSC niche cells is necessary for mating-induced GSC increase.

We next examined whether Mmp2 is necessary for OA-induced GSC increase. Our ex vivo culture experiment revealed that OA-induced GSC increase was suppressed in *c587 >Mmp2* RNAi flies, suggesting that Mmp2 acts downstream of OA signaling (*Figure 4C*). Moreover, the OA-dependent upregulation of pMad level in GSCs was suppressed in *c587 >Mmp2* RNAi flies (*Figure 4D*). Notably, in virgin females, *Mmp2* RNAi affected neither the GSC number nor the cap cell number, indicating that *Mmp2* RNAi does not influence the overall niche architecture (*Figure 4—figure supplement 1D*). Mmp2 in the mature follicle cells cleaves and downregulates collagen VI, also known as Viking (Vkg) and is the major component of the basement membrane (*Deady et al., 2017*; *Wang et al., 2008*). However, *Mmp2* RNAi had no effect on Vkg::GFP level around the cap cells (*Figure 4—figure supplement 1E*). Therefore, the suppression of mating-induced GSC increase in *Mmp2* RNAi does not likely depend on the collagen VI level.

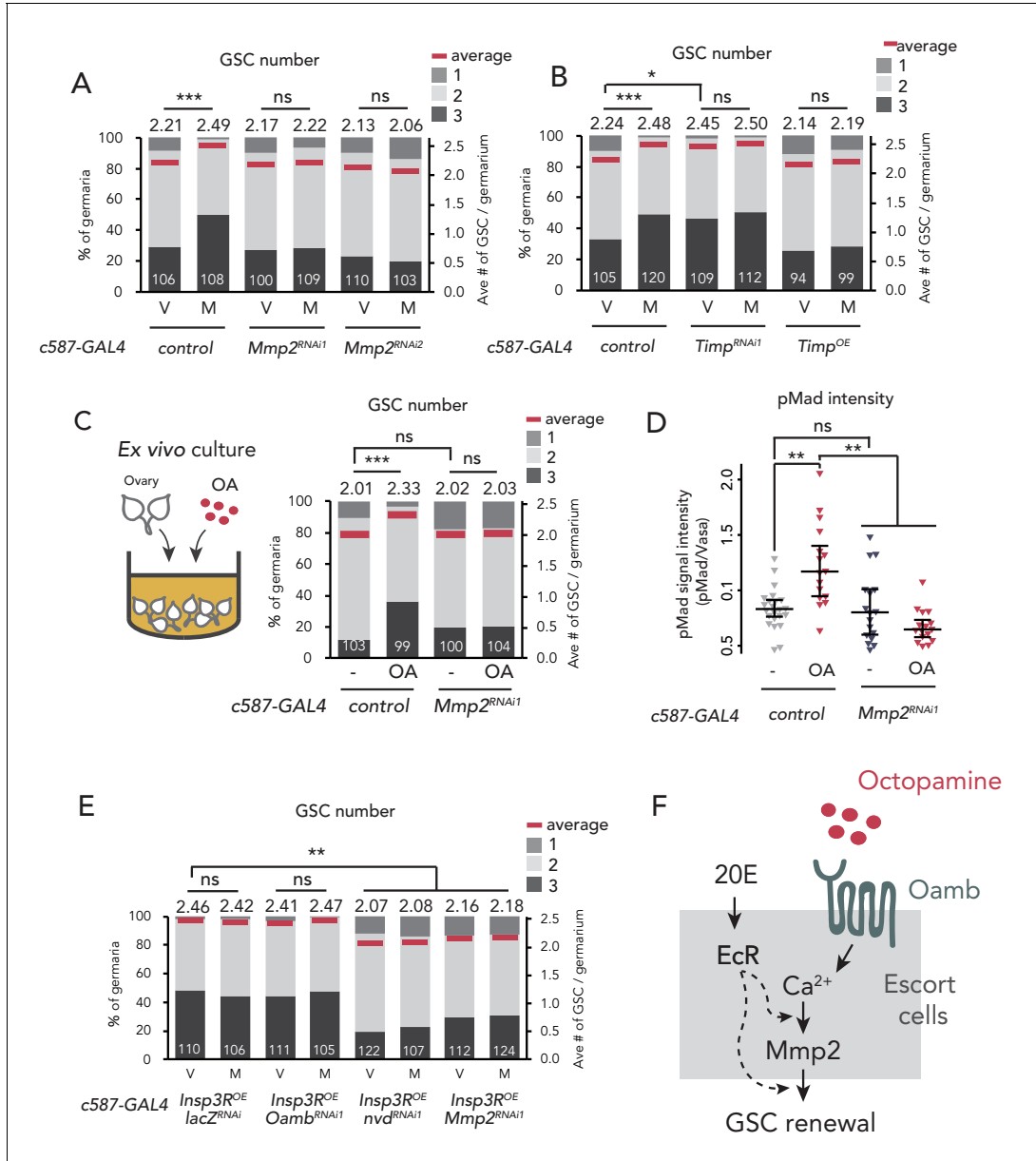

**Figure 4.** Mmp2 is necessary for OA-mediated GSC increase. (**A–C, E**) Frequencies of germaria containing 1, 2, and 3 GSCs (left vertical axis) and the average number of GSCs per germarium (right vertical axis) in virgin (**V**) and mated (**M**) female flies. *c587>+* flies were used as the control. The number of germaria analyzed is indicated inside the bars. (**A**) *Mmp2* RNAi by *c587-GAL4* driver. (**B**) RNAi and the overexpression of *Timp* by *c587-GAL4* driver. (**C**) Ex vivo culture experiment using *c587 >Mmp2* RNAi. OA was added into the ex vivo culture medium. Cultured with or without OA (+OA, -, respectively) is indicated under each bar. (**D**) Quantification of the relative pMad intensity in GSCs of the ex vivo cultured ovaries normalized to pMad intensity in CBs. Cultured with or without OA (+OA, −) is indicated under each bar. For the quantification of pMad intensity, the cell boundaries of GSCs and CBs were determined using anti-Vasa staining (n > 15). The three horizontal lines for each data sample indicate lower, median, and upper quartiles. (**E**) *Oamb*, *nvd*, or *Mmp2* RNAi in the genetic background of *c587 >Insp3R* overexpression. (**F**) A model of signaling in the escort cell to induce the mating-induced GSC increase. Oamb in the escort cells receives OA, and induce $[Ca^{2+}]_i$ in the cells. The $[Ca^{2+}]_i$ induces GSC increase via Mmp2. Ecdysteroid signaling is also involved in this process. Wilcoxon rank sum test with Holm's correction was used. ***$p \leq 0.001$, **$p \leq 0.01$, and *$p \leq 0.05$; NS, non-significant ($p > 0.05$). All source data are available in *Source data 1* and *2*.

The online version of this article includes the following figure supplement(s) for figure 4:

**Figure supplement 1.** Mmp2 is necessary in the escort cells to induce GSC increase.

To examine the epistasis of OA/Oamb-$Ca^{2+}$ signaling, ecdysteroid signaling, and Mmp2, we knocked down *Oamb*, *nvd*, or *Mmp2* in *c587 >Insp3R$^{OE}$* genetic background, where $Ca^{2+}$ signaling was forcedly upregulated. Whereas the *Oamb* RNAi did not suppress GSC increase in virgin female (*c587 >Insp3R$^{OE}$, Oamb$^{RNAi1}$*), the RNAi of *nvd* or *Mmp2* suppressed GSC increase even when $Ca^{2+}$ signaling were activated (*c587 >Insp3R$^{OE}$, nvd$^{RNAi}$* or *c587 >Insp3R$^{OE}$, Mmp2$^{RNAi1}$*) (*Figure 4E*). These results suggest that ecdysteroid signaling and Mmp2 act downstream of $Ca^{2+}$ signaling in OA-induced GSC increase. Taken together, both the OA-Oamb signaling and the downstream $Ca^{2+}$ signaling regulate the mating-induced GSC increase via Mmp2 and ecdysteroid signaling (*Figure 4F*).

## Octopamine from *dsx$^+$ Tdc2$^+$* neurons regulates the mating-induced GSC increase

To examine the in vivo role of OA in mating-induced GSC increase, we silenced the expression of *Tyrosine decarboxylase 2* (*Tdc2*) and *Tyramine β hydroxylase* (*TβH*) genes, which code for enzymes responsible for OA biosynthesis (*Cole et al., 2005*; *Monastirioti et al., 1996*; *Figure 5A*), with *Tdc2-GAL4* driver-mediated RNAi (*Tdc2 >Tdc2$^{RNAi1}$*, *Tdc2 >TbHRNAi$^{RNAi1}$*). Similar to the phenotype of *Oamb* RNAi, *Tdc2* or *TbH* RNAi with *Tdc2-GAL4* or *nSyb-GAL4* impaired the mating-induced GSC increase (*Figure 5A* and *Figure 5—figure supplement 1A–B*). Moreover, the impairment of GSC increase was restored when *Tdc2* or *TbH* RNAi flies were fed with food supplemented with OA (*Figure 5A*), supporting our hypothesis that OA is responsible for the mating-induced GSC increase in vivo.

Because *Tdc2-GAL4* is active in the nervous system (*Busch et al., 2009*; *Pauls et al., 2018*), we then identified which neurons secrete OA to the escort cells. *D. melanogaster* has more than 70–100 OAergic neurons dispersed throughout the nervous system (*Monastirioti, 2003*; *Schwaerzel et al., 2003*; *Zhou et al., 2008*). Among them, we were particularly interested in a small subset innervating the reproductive system (*Figure 5B*) as several recent studies have revealed that these neurons regulate mating behavior, egg laying, and ovarian-muscle contraction (*Heifetz et al., 2014*; *Lee et al., 2003*; *Middleton et al., 2006*; *Rezával et al., 2014*; *Rubinstein and Wolfner, 2013*). The ovary-projecting OAergic neurons are *doublesex* (*dsx*)$^+$ and *Tdc2$^+$* double-positive (*Rezával et al., 2014*; *Figure 5B*). Therefore, to manipulate the gene expression of *dsx$^+$ Tdc2$^+$* neurons only, we implemented a FLP/FRT intersectional strategy using *dsx-FLP* (*Rezával et al., 2014*). We could detect *GFP* expression only in the *dsx$^+$ Tdc2$^+$* neurons innervating to the ovary, whose cell bodies are located on a caudal part of the abdominal ganglion (*Rezával et al., 2014*; *Figure 5—figure supplement 1C–E*). We next knocked down *Tdc2* in *dsx$^+$ Tdc2$^+$* neurons by RNAi (*tub >GAL80>Tdc2$^{RNAi}$*; *dsx-FLP*) and found that these RNAi flies failed to increase GSC number after mating. Given the fact that the intersectional strategy using *dsx-FLP* does not label any neurons in the central nervous system (*Rezával et al., 2014*), these data suggest that only a small subset of *dsx$^+$ Tdc2$^+$* neurons controls the mating-induced GSC increase.

To assess whether the activity of *dsx$^+$ Tdc2$^+$* neurons affects the GSC number, we overexpressed *TrpA1*, a temperature-sensitive cation channel gene, in the *dsx$^+$ Tdc2$^+$* neurons only. In *Tdc2 >stop >TrpA1;dsx-FLP* flies, we can tightly control the *TrpA1* expression in the *dsx$^+$ Tdc2$^+$* neurons only (*Rezával et al., 2014*). Both the control flies and *TrpA1*-overexpressing flies at permissive temperature (17°C) had the normal GSC number in virgin and mated females. On the other hand, at the restrictive temperature (29°C), the *TrpA1*-overexpressing flies, even the virgin ones, had more GSCs (*Figure 5D*). We also found that *Tdc2 >stop >TrpA1; dsx-FLP* virgin females at the restrictive temperature had increased GSC frequency in the M phase (*Figure 5E*). Importantly, the *TrpA1*-mediated activation of *Tdc2* neurons did not induce the GSC increase in loss-of -*Oamb*-function females (*Tdc2 >TrpA1; Oamb$^\Delta$/Oamb$^\Delta$*) (*Figure 5F*), suggesting that the *TrpA1*-mediated GSC increase requires Oamb.

Furthermore, we employed the *Tetanus toxin light chain* (*TNT*) to inhibit neuronal activity (*Sweeney et al., 1995*). When we overexpressed *TNT* in *dsx$^+$ Tdc2$^+$* neurons only, the mating-induced GSC increase was suppressed in mated females as compared with the control, whose inactivated *TNT$^{in}$* was overexpressed (*Figure 5G*). Taken together, these findings suggest that the mating-induced GSC increase is mediated by the neuronal activity of *dsx$^+$ Tdc2$^+$* neurons innervating to the ovary.

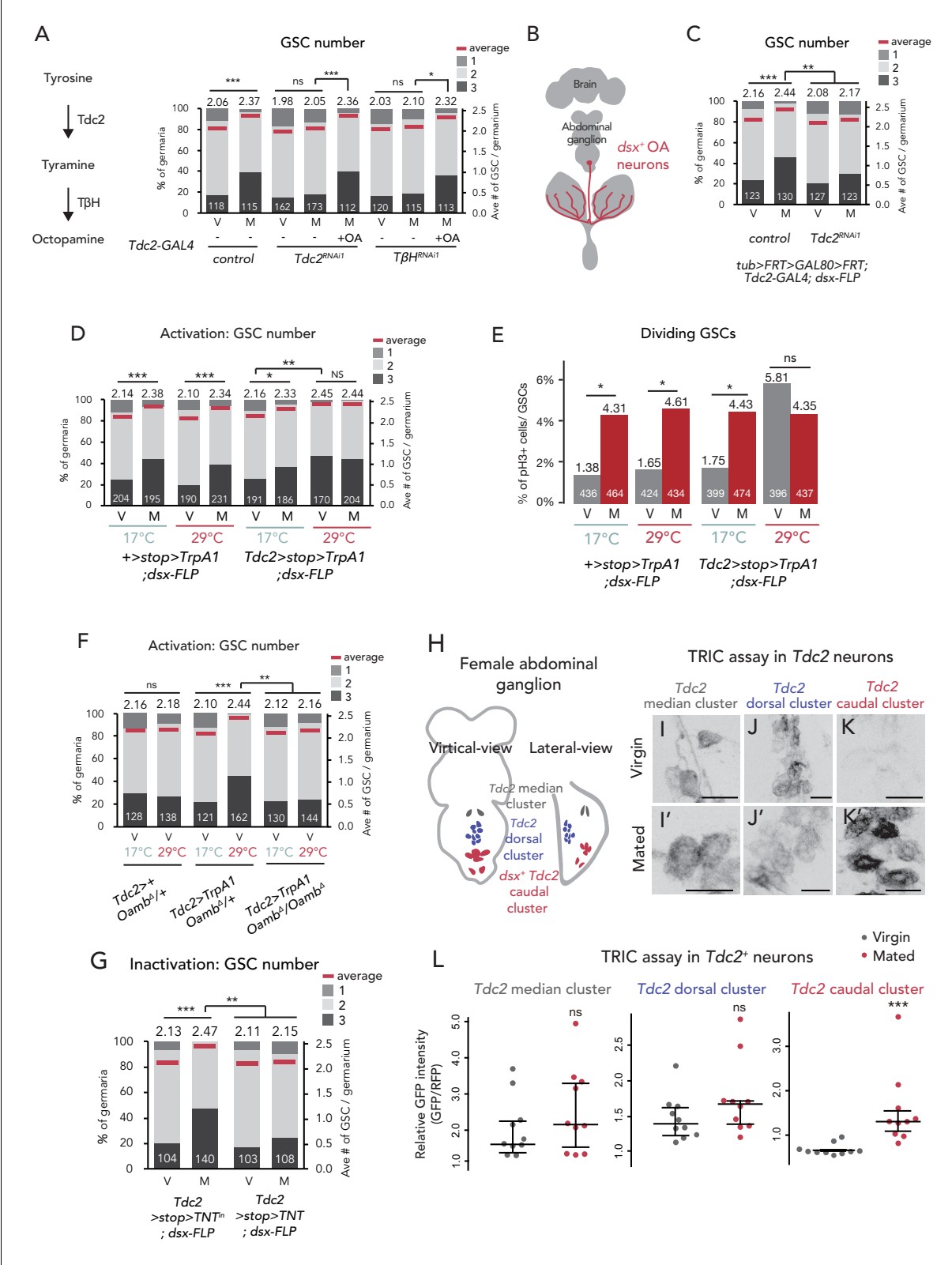

**Figure 5.** Ovary-projecting OA neurons control the GSC increase. (A, C–D, F–G) Frequencies of germaria containing 1, 2, and 3 GSCs (left vertical axis) and the average number of GSCs per germarium (right vertical axis) in virgin (V) and mated (M) female flies. The number of germaria analyzed is indicated inside the bars. (A) RNAi of *Tdc2* and *TβH* by *Tdc2-GAL4*. OA was added into the standard food. (B) A schematic drawing of *Drosophila* central nervous system and the ovary-projecting OA neurons with the *dsx*⁺ OA neurons projecting to the ovary. (C) *Tdc2* RNAi in *dsx*⁺ *Tdc2*⁺ neurons

*Figure 5 continued*

with the genotype indicated. (D–E) *TrpA1*-mediated activation of $dsx^+$ $Tdc2^+$ neurons. 17°C and 29°C were used as the permissive and restrictive temperatures, respectively, of TrpA1 channel. (D) GSC number. (E) The ratio of $pH3^+$ GSCs and total GSCs. (F) The activation of $Tdc2^+$ neurons with $Oamb^\Delta$ genetic background. (G) The inactivation of $dsx^+$ $Tdc2^+$ neurons. (H) Illustration showing the location of three clusters of $Tdc2^+$ neurons in the caudal part of the abdominal ganglion (I–K, I'–K'). Negative images of TRIC labeling (anti-GFP) in the abdominal ganglions of virgin (I–K) and mated females (I'–K') of TRIC (*Tdc2 >UAS-mCD8::RFP, UAS-p65AD::CaM LexAop2-mCD8::GFP; nSyb-MKII::nlsLexADBDo;UAS-p65AD::CaM*) flies, indicating intracellular $Ca^{2+}$ transients. Scale bars, 20 μm. (L) The GFP intensities from the $Tdc2^+$ median cluster, $Tdc2^+$ dorsal cluster, and $dsx^+$ $Tdc2^+$ cluster of TRIC females show $Ca^{2+}$ activity in virgin (gray) and mated females (red). Wilcoxon rank sum test was used for A, C, D, F, G, and L. Fisher's exact test with Holm's correction was used for E. ***$p \leq 0.001$, **$p \leq 0.01$, and *$p \leq 0.05$; NS, non-significant ($p > 0.05$). All source data are available in *Source data 1* and *3*.

The online version of this article includes the following figure supplement(s) for figure 5:

**Figure supplement 1.** $dsx^+$ $Tdc2^+$ neurons control GSC increase.

## $dsx^+$ $Tdc2^+$ neurons are activated after mating

Because the $dsx^+$ $Tdc2^+$ neuronal activity has a significant role in mating-induced GSC increase, we next examined whether these neurons change their activity before and after mating. We monitored the neuronal activity using an end-point $Ca^{2+}$ reporting system, the transcriptional reporter of intracellular $Ca^{2+}$ (TRIC) (*Gao et al., 2015*). TRIC is designed to increase the *GFP* expression in proportion to $[Ca^{2+}]_i$. We classified female $Tdc2^+$ neurons in the caudal part of the abdominal ganglion into three clusters based on their location and morphology. We designated the three clusters of these $Tdc2^+$ neurons as the $Tdc2^+$ median, $Tdc2^+$ dorsal, and $Tdc2^+$ caudal clusters (*Figure 5H*). Among them, the position of first two clusters are not similar to that of $dsx^+$ $Tdc2^+$ neurons, whereas that of the $Tdc2^+$ caudal cluster is similar (*Rezával et al., 2014*). In virgin females, we detected robust TRIC signals in the $Tdc2^+$ median and $Tdc2^+$ dorsal clusters but not in the $Tdc2^+$ caudal cluster (*Figure 5I–L*). In contrast, 24 hr after mating, we observed a significant increase in the TRIC signal in the $Tdc2^+$ caudal cluster, whereas those in the $Tdc2^+$ median and $Tdc2^+$ dorsal clusters were not changed in virgin and mated females (*Figure 5I–L*). This result suggests that the $Tdc2^+$ caudal cluster, which are likely $dsx^+$ $Tdc2^+$ neurons, is significantly activated after mating.

## The activity switch of $dsx^+$ $Tdc2^+$ neurons is regulated by the sex peptide sensory neurons via acetylcholine signaling

Our previous study revealed that the mating-induced GSC increase is mediated by the male seminal fluid protein SP (*Ameku and Niwa, 2016*). SP is received by SPR in a small number of SPSNs, followed by a neural silencing of SPSNs (*Häsemeyer et al., 2009*; *Yapici et al., 2008*). Notably, SPSNs project their arbors into a caudal part of the abdominal ganglion, where the cell bodies of the $dsx^+$ $Tdc2^+$ cluster neurons are located (*Rezával et al., 2014*; *Rezával et al., 2012*). Therefore, we examined whether SPSNs physically interact with $Tdc2^+$ neurons in the abdominal ganglion by performing the GFP Reconstitution Across Synaptic Partners (GRASP) analysis (*Feinberg et al., 2008*; *Gordon and Scott, 2009*), in which two complementary fragments of *GFP* were expressed in SPSNs and $Tdc2^+$ neurons. GRASP signals were detected in the abdominal ganglion (*Figure 6A*), suggesting that the axon termini of SPSNs and the cell bodies and/or dendrites of $Tdc2^+$ neurons contact each other likely through synaptic connections.

Because SPSNs have been implied as cholinergic neurons (*Rezával et al., 2012*), we next examined the expression of *Choline acetyltransferase* (*ChaT*)-GAL4 in SPSNs. *ChaT* encodes an acetylcholine biogenic enzyme (*Greenspan, 1980*). The SPSNs located on the oviduct, which also co-express *pickpocket* (*ppk*) and *fruitless* (*fru*), are particularly crucial for inducing the major behavioral changes in female flies after mating (*Ameku and Niwa, 2016*; *Rezával et al., 2012*). By using *UAS-mCD8::RFP* with *ChaT-GAL4* alongside *ppk-EGFP*, we confirmed that the *ppk-EGFP*–positive population near the oviduct were co-labeled by RFP (*Figure 6B*), consistent with the speculation that SPSNs are cholinergic.

We next counted the GSC number in *ChAT* RNAi flies using *ppk-GAL4*. *ppk >ChAT*[RNAi] virgin flies had more GSCs compared with the control (*Figure 6C*). In addition, mating did not induce GSC increase in *ppk >ChAT*[RNAi] flies, suggesting that the acetylcholine released from SPSNs is responsible for suppressing the GSC increase.

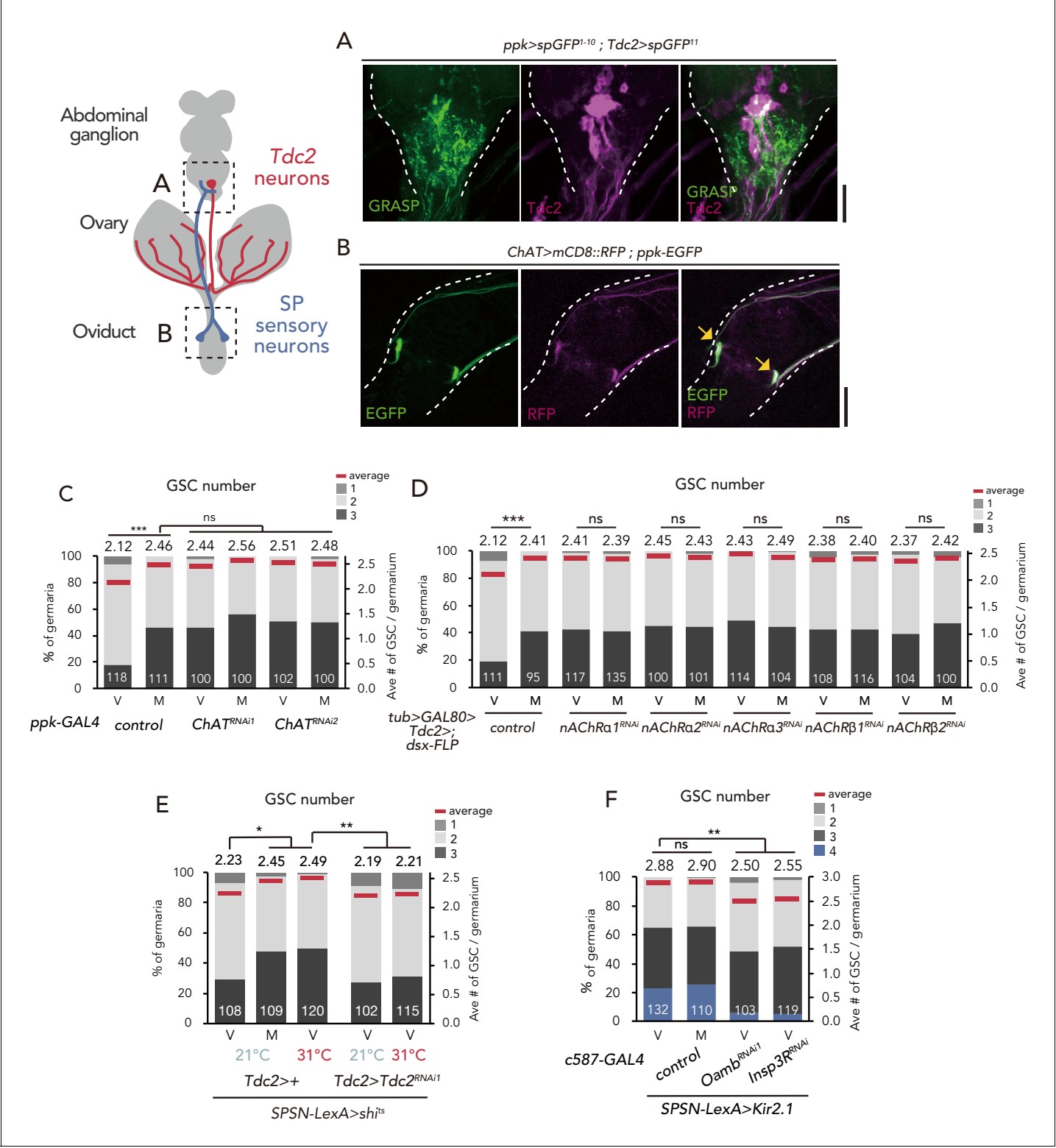

**Figure 6.** SPSNs control GSC increase through OA neurons. (**A**) Neuronal proximity of SPSNs and *Tdc2*[+] neurons in the abdominal ganglion of female flies stained with anti-Tdc2 (magenta). Note that reconstituted GFP (GRASP) signal was detected in the caudal part of the abdominal ganglion surrounded by broken white lines. Scale bar, 25 µm. (**B**) Cell bodies of SPSNs (yellow arrows) of *ChAT-GAL4; UAS-mCD8::RFP; ppk-EGFP* virgin females. Note that mCD8::RFP and EGFP signals overlapped in the cell bodies (yellow arrow) of SPSNs. White broken lines outline the oviduct. Scale bar, 25 µm. (**C–F**) Frequencies of germaria containing 1, 2, 3, and 4 GSCs (left vertical axis) and the average number of GSCs per germarium (right vertical axis) in
*Figure 6 continued on next page*

Figure 6 continued

virgin (V) and mated (M) female flies. The number of germaria analyzed is indicated inside the bars. (C) *ChaT* RNAi by *ppk-GAL4*. (D) RNAi of *nAChR*s in *dsx⁺ Tdc2⁺* neurons. (E) RNAi of *Tdc2* by *Tdc2-GAL4* along with the silencing of SPSNs. 21 and 31°C were used as the permissive and restrictive temperatures, respectively, of *shibire^ts (shi^ts)*. (F) RNAi of *Oamb* and *Insp3R* by *c587-GAL4* along with the silencing of SPSNs. *Kir2.1* was used in this experiment. Note that frequencies of germaria containing 4 GSCs increased. Wilcoxon rank sum test with Holm's correction was used for C, D, E and F. ***p≤0.001 and **p≤0.01; NS, non-significant (p>0.05). All source data are available in *Source data 1*.

The online version of this article includes the following figure supplement(s) for figure 6:

**Figure supplement 1.** *nAChRs* are expressed in the ovary-projecting *Tdc2* neurons.

**Figure supplement 2.** nAChRα1 in the *Tdc2* neurons regulates GSC increase.

To further ascertain whether the acetylcholine released from SPSNs is received by *dsx⁺ Tdc2⁺* neurons to mediate mating-induced GSC increase, we focused on the fast-ionotropic nicotinic acetylcholine receptors (nAChR), which belong to the Cys-loop receptor subfamily of ligand-gated ion channels (*Breer and Sattelle, 1987*; *Gundelfinger and Hess, 1992*; *Lee and O'Dowd, 1999*). In *D. melanogaster*, 10 genes coding *nAChR subunits* have been identified. Among these, we focused on *nAChRα1, nAChRα2, nAChRα3, nAChRβ1* and *nAChRβ2* because the knock-down of these genes in *dsx⁺ Tdc2⁺* (*tub >GAL80>, dsx-FLP; Tdc2-GAL4*) or *Tdc2* neurons (*Tdc2-GAL4*) increased the GSC number in virgin females similar to *ppk >ChaT^RNAi* (*Figure 6D* and *Figure 6—figure supplement 1A*). We then confirmed the expression of these *acetylcholine receptor* genes in *Tdc2* neurons by generating a knock-in *T2A-GAL4* line as previously described (*Kondo et al., 2020*; *Ihara et al., 2020*) for each 5 *nAChR subunit*s and observed their expression with *UAS-mCD8::GFP*. All of the five *knock-in-GAL4* expressions were detected in anti-Tdc2 positive neurons around the ovary, suggesting that the ovary-projecting *dsx⁺ Tdc2⁺* neurons expresses these *nAChRs* (*Figure 6—figure supplement 1B–F*).

To confirm the role of nAChR in mating-induced GSC increase, we generated *nAChRα1* complete loss-of-function genetic alleles by CRISPR/Cas9 technology (*Kondo and Ueda, 2013*; *Figure 6—figure supplement 2A*). Similar to *nAChRα1* RNAi females, the *nAChRα1* transheterozygous mutant virgin females (*nAChRα1²²⁸/nAChRα³²⁶*) had more GSCs compared with the controls (*Figure 6—figure supplement 2B*). In addition, the GSC increase of *nAChRα1²²⁸/nAChRα³²⁶* was restored by the over-expression of *nAChRa1* in *Tdc2⁺* neurons (*Tdc2 >nAChRα1; nAChRα1²²⁸/nAChRα³²⁶*) (*Figure 6—figure supplement 2C*). These data support our hypothesis that acetylcholine signaling in *Tdc2* neurons has a negative role in mating-induced GSC increase.

We next assessed relationship between SPSNs, *Tdc2⁺* neurons, and OA-Oamb-Ca²⁺ signaling in ovarian cells. The silencing of SPSNs neuronal activity (*SPSNs-LexA* and *LexAop-shi^ts*) increased the GSC number in virgin females (*Figure 6E*), consistent with our previous study (*Ameku and Niwa, 2016*). Upon SPSNs silencing, *Tdc2* RNAi by *Tdc2-GAL4* reduced the GSC number (*Figure 6E*), suggesting that *Tdc2⁺* neurons act downstream of SPSNs. In addition, the GSC increase through the silencing of SPSNs (*SPSNs-LexA >LexAop-kir2.1*) was suppressed by *Oamb* or *Insp3R* RNAi in the escort cells (*Figure 6F*), suggesting that OA-Oamb-Ca²⁺ signaling in ovarian cells acts downstream of SPSNs. Overall, our findings revealed a novel neuronal relay in response to mating that regulates the female GSC increase in the ovary before/after mating (*Figure 7*).

## Discussion

In this study, we report that the mating-induced GSC increase in female *D. melanogaster* is regulated by OAergic neurons directly projecting to the ovary. From our in vivo and ex vivo experiments, we propose the following model to explain the mating-induced GSC increase. After mating, the male seminal fluid SP is transferred into the female uterus, stimulating SPR-positive neurons. As the liganded SPR silences the neuronal activity of SPSNs (*Häsemeyer et al., 2009*), the acetylcholine released from SPSNs is suppressed. As SPSNs and *dsx⁺ Tdc2⁺* neurons are directly connected, this suppression directly modulates *dsx⁺ Tdc2⁺* neuronal activity. Because we have shown that nAChRs in *dsx⁺ Tdc2⁺* neurons exhibit an inhibitory effect with an unknown mechanism (to be discussed later), the inactivation of nAChRs in the absence of acetylcholine results in the activation of *dsx⁺ Tdc2⁺* neurons in mated females. As a consequence, OA is released from *dsx⁺ Tdc2⁺* neurons, received by Oamb, induces [Ca²⁺]ᵢ in the escort cells, and finally activates the Mmp2 enzymatic

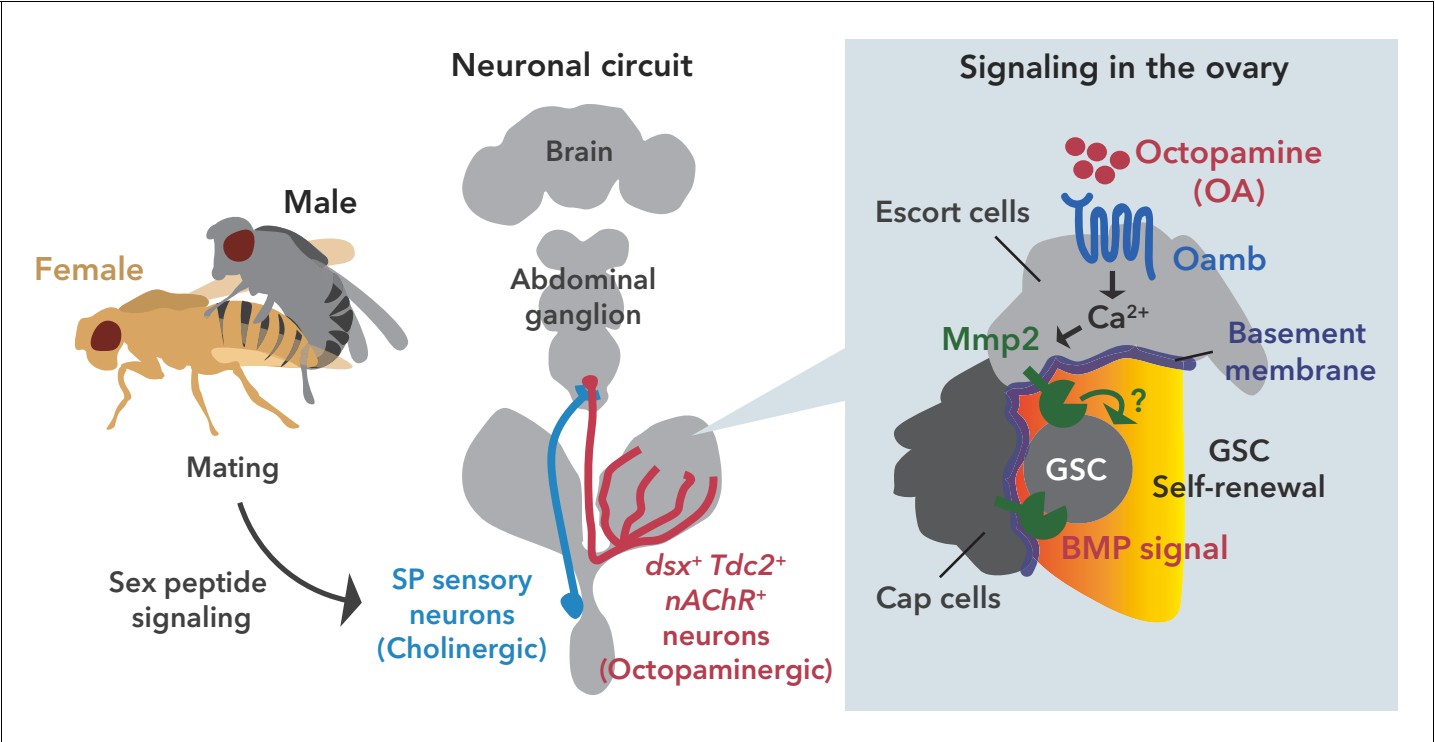

**Figure 7.** Neuronal octopamine signaling, followed by Oamb-Ca$^{2+}$-Mmp2 signaling, regulates the mating-induced GSC increase. The illustration is the proposed working model from our findings here. SP signaling and SP sensory neurons activate $dsx^+$ $Tdc2$ neurons via acetylcholine signaling. The octopamine released from $dsx^+$ $Tdc2$ neurons is received by the Oamb in escort cells and then activates intracellular Ca$^{2+}$ flux. The OA-mediated signaling increases pMad levels in GSCs to evoke mating-induced GSC increase via Mmp2.

activity. The activity of Mmp2 positively regulates the Dpp-mediated niche signaling, thereby leading to mating-induced GSC increase (*Figure 7*).

## The octopamine- and its receptor-dependent GSC control

Our proposed model is that the OA from $dsx^+$ $Tdc2^+$ neurons is directly received by the escort and follicle cells in the germarium. Our model is supported by two of our observations. First, mating-induced GSC increase is impaired by *Oamb* RNAi using a *GAL4* driver that is active specifically in the germarium cells but not mature follicle cells. Second, OA treatment evokes [Ca$^{2+}$]$_i$ elevation in these germarium cells in an Oamb-dependent manner. However, in this study, we did not address whether the escort cells and/or the follicle cells in the germarium express *Oamb*, as we failed to observe any clear *GAL4* expression in two independent *Oamb-T2A-GAL4* drivers (*Figure 1—figure supplement 2A,B,C and D*). We surmise that this may be due to lower amounts of *Oamb* transcript in the germarium.

We have shown that the activation of the ovary-projecting $dsx^+$ $Tdc2^+$ neurons is necessary and sufficient to induce GSC increase. However, from an anatomical point of view, the $dsx^+$ $Tdc2^+$ neurons project to the distal half of the ovary but not to the germarium (*Figure 5—figure supplement 1E*). Considering our model described above, this disagreement can be attributed to the characteristic volume transmission of monoamine neurotransmitters. In other words, neurotransmitters act at a distance well beyond their release sites from cells or synapses (*Fuxe et al., 2010*). Therefore, the OA secreted from the terminals of $dsx^+$ $Tdc2^+$ neurons could reach the germarium located at the most proximal part of the ovary.

Several previous studies have revealed that OA signaling has a pivotal role in reproductive tissues other than germarium, such as mature follicle cells, oviduct, and ovarian muscle, to promote ovulation, oviduct remodeling, and ovarian-muscle contraction, respectively (*Deady and Sun, 2015*; *Heifetz et al., 2014*; *Lee et al., 2009*; *Middleton et al., 2006*; *Rezával et al., 2014*). Therefore, it is likely that the $dsx^+$ $Tdc2^+$ neurons orchestrate multiple different events during oogenesis in

response to mating stimulus. Because a mated female needs to activate oogenesis to continuously produce eggs in concert with sperm availability, it is reasonable that the ovary-projecting neurons switch on the activity of the entire process of reproduction.

## The role of Mmp2 in mating-induced GSC increase

Based on our present study and several previous studies (*Deady et al., 2015*; *Deady and Sun, 2015*; *Knapp and Sun, 2017*), the OA-Oamb-$Ca^{2+}$-Mmp2 axis is required for GSC increase and follicle rupture, both of which are induced by mating stimuli in *D. melanogaster*. In both cases, Mmp2 enzymatic activity is likely to be essential, as the overexpression of *Timp* encoding a protein inhibitor of Mmp2 suppresses GSC increase, as well as follicle rupture. Mmp2 in mature follicle cells cleaves and downregulates Viking/collagen VI (*Deady et al., 2017*; *Wang et al., 2008*). In fact, several previous studies have revealed that Viking/collagen VI is required for GSC maintenance in female *D. melanogaster* (*Van De Bor et al., 2015*; *Wang et al., 2008*). However, we observed no significant change in Viking/Collagen VI levels in the germarium between the control and *Mmp2* RNAi flies (*Figure 4—figure supplement 1E*). Therefore, we concluded that Viking/collagen VI is not a substrate of Mmp2 in the regulation of mating-induced GSC increase. Besides Viking/Collagen VI, Dally-like (Dlp) is another basement membrane protein associated with extracellular matrix and known as the Mmp2 substrate (*Wang and Page-McCaw, 2014*). Interestingly, *dlp* is expressed in the escort cells (*Wang and Page-McCaw, 2014*). Moreover, Dlp controls the distribution of Dpp and Wnts, both of which significantly affect GSC self-renewal and differentiation (*Wang et al., 2015*; *Xie and Spradling, 1998*). Future research should decipher the exact substrate by which Mmp2 controls Dpp and/or Wnts to modulate GSC behavior in response to mating stimulus.

Another remaining question to be addressed is how Mmp2 function is regulated in GSC increase. Ecdysteroid biosynthesis and signaling in the ovary are necessary but not sufficient for the OA-Oamb-$Ca^{2+}$–mediated GSC increase and follicle rupture (*Ameku and Niwa, 2016*; *Knapp and Sun, 2017*). We found that in the regulation of mating-induced GSC increase, ecdysteroid signaling acts downstream of $Ca^{2+}$ signaling (*Figure 4F*). On the other hand, in the follicle rupture process, ecdysteroid signaling either acts downstream, upstream, or both, of $Ca^{2+}$ signaling. Further, the precise action of ecdysteroid has yet to be elucidated (*Knapp and Sun, 2017*). The Mmp2-GFP fusion protein level in the follicle cells is not changed in the loss-of-*Ecdysone receptor*-function flies, implying that ecdysteroid signaling might regulate Mmp2 enzymatic activity by an unknown mechanism (*Knapp and Sun, 2017*). Considering the involvement of both the OA-Oamb-$Ca^{2+}$-Mmp2 axis and ecdysteroid biosynthesis, it is very likely that the Mmp2 enzymatic activity is also regulated by the same, unknown mechanism not only in the mature follicle cells to control follicle rupture, but also in the germarium to control mating-induced GSC increase.

## SPSN-mediated suppression of *dsx⁺/Tdc2⁺* neurons

In many animals, reproduction involves significant behavioral and physiological shifts in response to mating. In female *D. melanogaster*, several post-mating responses are coordinated by SPSNs and their downstream afferent neuronal circuit (*Wang et al., 2020*), including Stato-Acoustic Ganglion neurons, the ventral abdominal lateral Myoinhibitory peptide neurons, and the efferent *dsx⁺ Tdc2⁺* neurons (*Feng et al., 2014*; *Häsemeyer et al., 2009*; *Jang et al., 2017*; *Rezával et al., 2014*). Our GRASP analysis indicates a direct synaptic connection between cholinergic SPSNs and OAergic neurons. Moreover, we demonstrated that nAChRs in *dsx⁺ Tdc2⁺* neurons are responsible for the suppression of their neuronal activity in virgin females. However, nAChRs are the cation channels leading to depolarization upon acetylcholine binding, and therefore usually activate neurons (*Corringer et al., 2000*; *Lee and O'Dowd, 1999*; *Perry et al., 2012*). How is the opposite role of nAChRs in *dsx⁺ Tdc2⁺* neuronal activity achieved? One possibility is that acetylcholine-nAChR signaling does not evoke a simple depolarization but rather generates a virgin-specific temporal spike pattern in *dsx⁺ Tdc2⁺* neurons. Interestingly, recent studies demonstrated that the pattern, instead of the frequency, of neuronal firing is significant in adjusting the neuronal activity of clock neurons in *D. melanogaster* (*Tabuchi et al., 2018*). The firing pattern relies on control of ionic flux by the modulation of $Ca^{2+}$-activated potassium channel and $Na^+/K^+$ ATPase activity. Because whether mating changes the firing pattern of *dsx⁺ Tdc2⁺* neurons remains to be examined, the neuronal activity in

SPSNs and the *dsx⁺ Tdc2⁺* neuronal circuit between virgin and mated females are future research areas.

## Interorgan communication among multiple organs to regulate the increase and maintenance of female GSCs

In the last decades, there is growing evidence that GSCs and their niche are influenced by multiple humoral factors (*Drummond-Barbosa, 2019*; *Yoshinari et al., 2019*). Based on the data from our current study and previous studies, there are at least four crucial humoral factors for regulating the increase and/or maintenance of *D. melanogaster* female GSCs, including DILPs (*Hsu et al., 2008*; *Hsu and Drummond-Barbosa, 2009*; *LaFever, 2005*), ecdysteroids (*Ables and Drummond-Barbosa, 2010*; *Ameku et al., 2017*; *Ameku and Niwa, 2016*; *König et al., 2011*), Neuropeptide F (NPF) (*Ameku et al., 2018*), and OA (this study). Notably, all of these come from different sources: DILPs are from the insulin-producing cells located in the pars intercerebralis of the central brain; ecdysteroids from the ovary; NPF from the midgut; and OA from the neurons located in the abdominal ganglion. In addition to these identified humoral factors, recent studies also imply that adiponectin and unknown adipocyte-derived factor(s) are essential for GSC maintenance (*Armstrong and Drummond-Barbosa, 2018*; *Laws et al., 2015*; *Matsuoka et al., 2017*). These data clearly indicate that *D. melanogaster* female GSCs are systemically regulated by interorgan communication involving multiple organs. The additional interorgan communication mechanisms that ensure the faithful coupling of the increase and maintenance of GSC to the organism's external and physiological environments are essential to be investigated in future studies.

To modulate the increase and maintenance of GSC, ecdysteroids are received by both GSCs and niche cells (*Ables and Drummond-Barbosa, 2010*; *König et al., 2011*), whereas DILPs, NPF, and OA are received by niche cells. A major signal transduction mechanism of each of these humoral factors have been well characterized, namely phosphoinositide 3-kinase pathway for DILPs-InR signaling, EcR/Ultraspiracle-mediated pathway for ecdysteroid signaling, cAMP pathway for NPF-NPFR signaling (*Garczynski et al., 2002*), and $Ca^{2+}$ pathway for OA-Oamb signaling. However, it remains unclear whether and how each of these signaling pathways control the production and secretion of the niche signal, as well as its distribution and transduction. In addition, it is important to understand whether and how the multiple system signals are integrated to control the mating-induced increase and maintenance of GSCs.

## Evolutionarily conservation of monoamine-steroid hormone axis to control female reproduction

In recent years, many studies have revealed that not only local niche signals but also systemic and neuronal factors play indispensable roles in regulating GSC behavior (*Ables and Drummond-Barbosa, 2017*; *Drummond-Barbosa, 2019*; *Yoshinari et al., 2019*). In *D. melanogaster*, ecdysteroid signaling is essential for the proliferation and maintenance of GSCs and neural stem cells (*Ables and Drummond-Barbosa, 2010*; *Homem et al., 2014*; *König et al., 2011*). In this study, we have identified the ovary-projecting OAergic neurons as new regulators of stem cell homeostasis. Both steroid hormones and OA-like monoamines, such as noradrenaline, are also involved in stem cell regulation in mammals. For example, the mammalian steroid hormone, estrogen, is important in regulating cell division and/or maintenance of hematopoietic stem cells, mammary stem cell, neural stem cells, and hematopoietic stem cells (*Asselin-Labat et al., 2010*; *Bramble et al., 2019*; *Kim et al., 2016*; *Nakada et al., 2014*). Moreover, noradrenergic neurons, which directly project to the bone marrow, regulate the remodeling of hematopoietic stem cells niche (*Ho et al., 2019*; *Méndez-Ferrer et al., 2010*; *Méndez-Ferrer et al., 2008*). Therefore, the steroid hormone- and noradrenergic nerve-dependent control of stem cell homeostasis are likely conserved across animal species. In this regard, the *D. melanogaster* reproductive system will further serve as a powerful model to unravel the conserved systemic and neuronal regulatory mechanisms for stem cell homeostasis in animals.

## Materials and methods

### *Drosophila* strains

Flies were raised on cornmeal-yeast-agar medium at 25°C. $EcR^{A483T}$, temperature-sensitive mutants, were cultured at 31°C for 1 d prior to the assays. $w^{1118}$ was used as the control strain.

The genetic mutant stocks used were $EcR^{A483T}$ (Bloomington Drosophila Stock Center [BDSC] #5799) and $EcR^{M554fs}$ (BDSC #4894). The protein-trap GFP line of *Vkg* (*Vkg::GFP*) was obtained from Kyoto Stock Center (DGRC #110692). *Dad-LacZ* (*Tsuneizumi et al., 1997*) (a gift from Yoshiki Hayashi, University of Tsukuba, Japan).

The following *GAL4* and *LexA* strains were used: *c587-GAL4* (*Manseau et al., 1997*) (gift from Hiroko Sano, Kurume University, Japan), *R44E10-GAL4* (*Deady and Sun, 2015*) (a gift from Jianjun Sun, University of Connecticut, USA), *RS-GAL4* (*Lee et al., 2009*) (a gift from Kyung-An Han, Pennsylvania State University, USA), *nSyb-GAL4* (BDSC #51941), *nSyb-GAL80* (*Harris et al., 2015*) (a gift from James W. Truman, Janelia Research Campus, USA), *tj-GAL4* (DGRC #104055), *R13C06-GAL4* (BDSC #47860), *109–30 GAL4* (BDSC #7023), *c355-GAL4* (BDSC #3750), *c306-GAL4* (BDSC #3743), *slbo-GAL4* (BDSC #6458), *bab1-GAL4* (*Bolívar et al., 2006*) (a gift from Satoru Kobayashi, University of Tsukuba, Japan), *nos-GAL4* (DGRC #107748), *tub >FRT >GAL80>FRT* (BDSC #38879), $Oamb^{KI-RD}$-*GAL4* (BDSC#84677) (*Deng et al., 2019*), *Oamb-KI-T2A-GAL4, nAChRα1-T2A-GAL4, nAChRα2-T2A-GAL4, nAChRα3-T2A-GAL4, nAChRβ1-T2A-GAL4, nAChRβ2-T2A-GAL4* (*Kondo et al., 2020*; *Ihara et al., 2020*), *ChaT-GAL4* (BDSC #6793), *ppk-GAL4* (*Grueber et al., 2007*) (a gift from Hiroko Sano, Kurume University, Japan), and *SPSNs-LexA* (*Feng et al., 2014*) (a gift from Young-Joon Kim, Gwangju Institute of Science and Technology, South Korea).

The following *UAS* and *LexOp* strains were used: *20xUAS-6xGFP* (BDSC #52261), *UAS-GFP; UAS-mCD8::GFP* (*Ito et al., 1997*; *Lee and Luo, 1999*) (a gift from Kei Ito, University of Cologne, Germany), *UAS-Stingar* (BDSC #84277), *UAS-mCD8::RFP* (BDSC #32219), *UAS-CsChrimson* (BDSC #55134), *UAS-Insp3R* (BDSC #30742), $UAS-Oamb^{AS}$ (*Lee et al., 2009*) (a gift from Kyung-An Han, Pennsylvania State University, USA), *UAS-Timp* (BDSC #58708) (a gift from Andrea Page-McCaw, Vanderbilt University, USA), $UAS > stop >dTrpA1^{mcherry}$, *UAS > stop >TNT*, $UAS > stop >TNT^{in}$ (*von Philipsborn et al., 2011*; *Yu et al., 2010*), *dsx-FLP* (*Rezával et al., 2014*) (a gift from Daisuke Yamamoto, Advanced ICT Research Institute, National Institute of Information and Communications Technology, Japan) TRiC; *UAS-mCD8::RFP, LexAop2-mCD8::GFP;nSyb-MKII::nlsLexADBDo;UAS-p65AD::CaM* (BDSC:61679), *ppk-eGFP* (*Grueber et al., 2003*) (a gift from Tadashi Uemura, Kyoto University, Japan), and *LexAop-Kir2.1* (*Feng et al., 2014*) (a gift from Young-Joon Kim, Gwangju Institute of Science and Technology, South Korea).

The RNAi transgenic lines used were as follows: $UAS-LacZ^{RNAi}$ (a gift from Masayuki Miura, The University of Tokyo, Japan), $UAS-Oamb^{RNAi1}$(BDSC #31171), $UAS-Oamb^{RNAi2}$(BDSC #31233), $UAS-Oamb^{RNAi3}$ (Vienna Drosophila Resource Center [VDRC] #106511), $UAS-Octβ1R^{RNAi}$(VDRC #110537), $UAS-Octβ2R^{RNAi}$(VDRC #104524), $UAS-Octβ3R^{RNAi}$ (VDRC #101189), $UAS-Insp3R^{RNAi}$ (BDSC #25937), $UAS-EcR^{RNAi}$ (VDRC #37059), $UAS-Mmp2^{RNAi1}$ (BDSC #31371), $UAS-Mmp2^{RNAi2}$ (VDRC #330303), $UAS-Timp^{RNAi1}$ (BDSC #61294), $UAS-Timp^{RNAi2}$ (VDRC #109427), $UAS-Tdc2^{RNAi1}$ (VDRC #330541), $UAS-Tdc2^{RNAi2}$ (BDSC #25871), $UAS-Tβh^{RNAi1}$ (VDRC #107070), $UAS-Tβh^{RNAi2}$ (BDSC #67968), $UAS-ChAT^{RNAi1}$ (VDRC #330291), $UAS-ChAT^{RNAi2}$ (BDSC #25856), $UAS-nAChRα1^{RNAi}$ (VDRC #48159), $UAS-nAChRα2^{RNAi}$ (VDRC #101760), $UAS-nAChRα3^{RNAi}$ (VDRC #101806), $UAS-nAChRβ1^{RNAi}$ (VDRC #106570), $UAS-nAChRβ2^{RNAi}$ (VDRC #109450), $UAS-nvd^{RNAi1}$, and $UAS-nvd^{RNAi2}$ (*Yoshiyama et al., 2006*).

### Generation of *Oamb* and *nAChRα1* genetic loss-of-function mutant strains

The mutant alleles $Oamb^{\Delta}$ (*Figure 1—figure supplement 1F*), $nAChRα1^{228}$, and $nAChRα1^{326}$ (*Figure 6—figure supplement 2A*) were created in a *white* (*w*) background using CRISPR/Cas9 as previously described (*Kondo and Ueda, 2013*). The following guide RNA (gRNA) sequences were used: *Oamb*, 5'-GATGAACTCGAGTACGGCCA-3', and 5'-GCGATCTCTGGTGCCGCATT-3'; $nAChRα1^{228}$, 5'-GGACATCATGCGTGTGCCGG-3'; $nAChRα1^{326}$, 5'-GGGCAGGTAGAAGACCAGAA-3'. The breakpoint detail of $Oamb^{\Delta}$ is described in *Figure 1—figure supplement 1F*, whereas those of $nAChRα1^{228}$ and $nAChRα1^{326}$ are described in *Figure 6—figure supplement 2A*.

### Generation of *UAS-nAChRα1* transgenic line

The pcDNA3.1 plasmid containing the wild-type *D. melanogaster* *nAChRα1* coding sequences (*nAChRα1*-pcDNA3.1) was synthesized previously described (*Ihara et al., 2018*). Briefly, *nAChRα1*-pcDNA3.1 was digested with *EcoR*I and *Not*I, and then the digested *nAChRα1* fragment was ligated with a *EcoR*I-*Not*I–digested pWALIUM10-moe plasmid (*Perkins et al., 2015*). Transformants were generated using the phiC31 integrase system in the P{CaryP}attP40 strain (*Groth et al., 2004*). The $w^+$ transformants of pWALIUM10-moe were established using standard protocols.

### Behavioral assays

Flies were reared at 25°C and aged for 5–6 d. Virgin female flies were mated overnight to $w^{1118}$ male flies at 25°C (10 males and 5–8 females per vial). For the thermal activation assays, flies were first reared at 17°C for 6 d and transferred to 29°C. In the case of *EcR* mutant assays, flies were transferred to 31°C for 24 hr before mating or ex vivo culture.

For OA feeding, newly eclosed virgin females were aged for 4 d in vials with standard food containing 7.5 mg/mL of OA (*Monastirioti et al., 1996*; *Rubinstein and Wolfner, 2013*).

### Immunohistochemistry

Tissues were dissected in phosphor buffer serine (PBS) and fixed in 4% paraformaldehyde in PBS for 30 to 60 min at room temperature (RT). The fixed samples were washed three times in PBS supplemented with 0.2% Triton X-100, blocked in blocking solution (PBS with 0.3% Triton X-100% and 0.2% bovine serum albumin [BSA]) for 1 hr at RT, and incubated with a primary antibody in the blocking solution at 4°C overnight. The primary antibodies used were chicken anti-GFP (Abcam #ab13970; 1:4,000), rabbit anti-RFP (Medical and Biological Laboratories PM005; 1:2,000), mouse anti-Hts 1B1 (Developmental Studies Hybridoma Bank [DSHB]; 1:50), rat anti-DE-cadherin DCAD2 (DSHB; 1:50), rabbit anti-pH3 (Merck Millipore #06–570; 1:1000), rabbit monoclonal anti-pMad (Abcam #ab52903; 1:1000), mouse anti-Lamin C LC28.26 (DSHB; 1:10), rabbit cleaved Dcp-1 (Cell Signaling Technology #9578; 1:100), rat anti-Vasa (DSHB; 1:50), mouse anti-LacZ (β-galactosidase) (DSHB#40-1a; 1:50), rabbit anti-Tdc2 (Abcam #ab128225; 1:2000), Alexa Fluor 546 phalloidin (Thermo Fisher Scientific #A22283; 1:200), and Alexa Fluor 633 phalloidin (Thermo Fisher Scientific #A22284; 1:200). After washing, fluorophore (Alexa Fluor 488, 546 or 633)-conjugated secondary antibodies (Thermo Fisher Scientific) were used at a 1:200 dilution, and the samples were incubated for 2 hr at RT in the blocking solution. After another washing step, all samples were mounted in FluorSave reagent (Merck Millipore #345789). GSC numbers were determined based on the morphology and position of their anteriorly anchored spherical spectrosome (*Ables and Drummond-Barbosa, 2010*; *Ameku et al., 2018*; *Ameku and Niwa, 2016*). Cap cells were identified by immunostaining with anti-Lamin C antibody as previously described (*Ables and Drummond-Barbosa, 2010*).

### Ex vivo ovary culture

We used 5–6-day-old females. The ovaries were dissected in Schneider's *Drosophila* medium (Thermo Fisher Scientific #21720024) and isolated from oviduct using forceps. Approximately 5–6 ovaries were immediately transferred to a dish containing 3 mL of Schneider's *Drosophila* medium supplemented with 15% fetal calf serum and 0.6% penicillin-streptomycin with/without the addition of OA (Sigma, final concentration of OA is 0–1000 µM) and 20E (Enzo Life Sciences; final concentration of 20 nM). The cultures were incubated at RT (except for *EcR* mutant flies, *Figure 3D*) for 16 hr, and the samples were immunostained to determine the GSC number.

### Ex vivo calcium imaging

We employed the previously described imaging methods to visualize GSC behavior (*Morris and Spradling, 2011*; *Reilein et al., 2018*). For the live imaging, the ovaries dissected from adult virgin female flies were placed on a glass bottom dish (IWAKI #4970–041) with 3 mL of Schneider's *Drosophila* medium and 100 µL of the test reagent (Schneider's *Drosophila* medium containing 300 mM OA) placed directly at the center of each dish. The images were obtained with a × 40 objective lens (water-immersion) using a Zeiss LSM 700 confocal microscope and were recorded every 4 s. The GCaMP6s fluorescence intensity in the escort cell was then calculated for each time point. The ratio of fluorescence (ΔF) at each time point was calculated by normalizing the fluorescence with the initial

fluorescence ($F_0$). The initial fluorescence ($F_0$) is the average GCaMP6s fluorescence intensity before adding the test reagent.

## Optogenetic activation of the escort cells

Red-shifted channelrhodopsin CsChrimson (*Klapoetke et al., 2014*) was used to increase the $[Ca^{2+}]_i$ in the escort cells by light. *UAS-CsChrimson* was expressed using *c587-GAL4* with *nSyb-GAL80*. All crosses and the early development of flies were performed under dark conditions. The experiment was done at 25°C. Adult flies were raised with standard food for 3 d after eclosion and then with standard food with 1 mM all-trans-retinal (ATR) for 3 d. Subsequently, they were kept in the presence of orange–red light from LED for 24 hr. LED light was shone from the outside of the plastic chamber covered by aluminum foil to enhance light intensity.

## Statistical analysis

All experiments were performed independently at least twice. Fluorescence intensity in confocal sections was measured via ImageJ. For pMad quantification, signal intensity was calculated by measuring the fluorescence intensity in GSCs and CBs, which were co-stained with anti-Vasa antibody to visualize their cell boundaries. Sample sizes were chosen based on the number of independent experiments required for statistical significance and technical feasibility. The experiments were not randomized, and the investigators were not blinded. All statistical analyses were carried out using the 'R' software environment. The P value is provided in comparison with the control and indicated as * for $p \leq 0.05$, ** for $p \leq 0.01$, *** for $p \leq 0.001$, and 'NS' for non-significant ($p > 0.05$).

## Acknowledgements

We thank Toshiro Aigaki, Kendal S Broadie, Aki Ejima, Kyung-An Han, Yukako Hattori, Yoshiki Hayashi, Makoto Ihara, Young-Joon Kim, Satoru Kobayashi, Kazuhiko Matsuda, Masayuki Miura, Akira Nakamura, Takashi Nishimura, Andrea Page-McCaw, Hiroko Sano, Jianjun Sun, Nobuaki Tanaka, James W Truman, Tadashi Uemura, Daisuke Yamamoto, the Bloomington Stock Center, the Kyoto Stock Center (DGRC), the National Institute of Genetics, the Vienna Drosophila Resource Center, and the Developmental Studies Hybridoma Bank for providing stocks and reagents; Aki Hori and Reiko Kise for their technical assistance; and Satoru Kobayashi and Shosei Yoshida for their helpful discussion. YY and TA were recipients of the fellowship from the Japan Society for the Promotion of Science. This work was supported by grants from KAKENHI (19H05240 to RN, 26250001 and 17H01378 to HT, 18J20572 to YY, and 15J00652 to TA), AMED-PRIME, AMED (17gm6010011h0001 to TK), AMED-CREST, AMED (19gm1110001h0003 to RN), and the Takeda Science Foundation to RN.

## Additional information

### Funding

| Funder | Grant reference number | Author |
|---|---|---|
| Japan Agency for Medical Research and Development | 19gm1110001h0003 | Ryusuke Niwa |
| Takeda Science Foundation | | Ryusuke Niwa |
| Japan Society for the Promotion of Science | KAKENHI 18J20572 | Yuto Yoshinari |
| Japan Society for the Promotion of Science | KAKENHI 15J00652 | Tomotsune Ameku |
| Japan Society for the Promotion of Science | KAKENHI 26250001 | Hiromu Tanimoto |
| Japan Society for the Promotion of Science | KAKENHI 17H01378 | Hiromu Tanimoto |
| Japan Agency for Medical Research and Development | 17gm6010011h0001 | Takayuki Kuraishi |

| Japan Society for the Promotion of Science | KAKENHI 19H05240 | Ryusuke Niwa |

The funders had no role in study design, data collection and interpretation, or the decision to submit the work for publication.

## Author contributions

Yuto Yoshinari, Conceptualization, Resources, Data curation, Formal analysis, Funding acquisition, Validation, Investigation, Visualization, Writing - original draft, Writing - review and editing; Tomotsune Ameku, Resources, Formal analysis, Funding acquisition, Investigation, Writing - review and editing; Shu Kondo, Yuko Shimada-Niwa, Resources, Validation, Investigation, Methodology, Writing - review and editing; Hiromu Tanimoto, Takayuki Kuraishi, Resources, Funding acquisition, Validation, Investigation, Methodology, Writing - review and editing; Ryusuke Niwa, Conceptualization, Resources, Data curation, Formal analysis, Supervision, Funding acquisition, Validation, Investigation, Methodology, Writing - original draft, Project administration, Writing - review and editing

## Author ORCIDs

Hiromu Tanimoto http://orcid.org/0000-0001-5880-6064
Yuko Shimada-Niwa http://orcid.org/0000-0001-5757-4329
Ryusuke Niwa https://orcid.org/0000-0002-1716-455X

## Decision letter and Author response

Decision letter https://doi.org/10.7554/eLife.57101.sa1
Author response https://doi.org/10.7554/eLife.57101.sa2

# Additional files

## Supplementary files

• Source data 1. Source data of numbers of GSC and other cells. Raw data of numbers of GSC and other cells for *Figure 1*, *Figure 1—figure supplement 1*, *Figure 1—figure supplement 3*, *Figure 2*, *Figure 2—figure supplement 1*, *Figure 3*, *Figure 4*, *Figure 4—figure supplement 1*, *Figure 5*, *Figure 5—figure supplement 1*, *Figure 6*, and *Figure 6—figure supplement 2*.

• Source data 2. Source data of pMad and Dad-lacZ signal intensity. Raw data of pMad and Dad-LacZ signal intensity for *Figure 1*, *Figure 1—figure supplement 3*, *Figure 2*, and *Figure 4*.

• Source data 3. Source data of the TRIC signal intensity. Raw data of the TRIC signal intensity for *Figure 5*.

• Source data 4. Source data of the GCaMPs signal intensity. Raw data of the GCaMPs signal intensity for *Figure 2*.

• Transparent reporting form

## Data availability

All data generated or analysed during this study are included in the manuscript and supporting files. Source data files have been provided for figures representing germline stem cell number, pMad signal intensity, GCaMP6 signal intensity, and TRIC signal intensity. Data has been deposited in Dryad under https://doi.org/10.5061/dryad.zkh189375.

The following dataset was generated:

| Author(s) | Year | Dataset title | Dataset URL | Database and Identifier |
|---|---|---|---|---|
| Niwa R, Yoshinari Y, Ameku T, Kondo S, Tanimoto H, Kuraishi T, Shimada-Niwa Y | 2020 | Data from: Neuronal octopamine signaling regulates mating-induced germline stem cell proliferation in female *Drosophila* | https://doi.org/10.5061/dryad.zkh189375 | Dryad Digital Repository, 10.5061/dryad.zkh189375 |

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

# Appendix 1

**Appendix 1—key resources table**

| Reagent type (species) or resource | Designation | Source or reference | Identifiers | Additional information |
|---|---|---|---|---|
| Genetic reagent (*D. melanogaster*) | c587-GAL4 | *Manseau et al., 1997* | FBal0150629 RRID:BDSC_67747 | A gift from Hiroko Sano, Kurume University, Japan |
| Genetic reagent (*D. melanogaster*) | Oamb$^\Delta$ | This paper | | Detail described in *Figure 1—figure supplement 1F* |
| Genetic reagent (*D. melanogaster*) | nAChRa1$^{228}$ | This paper | | Detail described in *Figure 6—figure supplement 2A* |
| Genetic reagent (*D. melanogaster*) | nAChRa1$^{326}$ | This paper | | Detail described in *Figure 6—figure supplement 2A* |
| Genetic reagent (*D. melanogaster*) | EcR$^{A483T}$ | Bloomington Drosophila Stock Center | BDSC: #5799 RRID:BDSC_5799 FBal0083501 | |
| Genetic reagent (*D. melanogaster*) | EcR$^{M554fs}$ | Bloomington Drosophila Stock Center | BDSC: #4894 RRID:BDSC_4894 FBal0083490 | |
| Genetic reagent (*D. melanogaster*) | Vkg::GFP | KYOTO stock center | DGRC #110692 RRID:DGGR_110692 FBal0286156 | |
| Genetic reagent (*D. melanogaster*) | Dad-LacZ | *Tsuneizumi et al., 1997* | FBal0065787 RRID:DGGR_118114 | A gift from Yoshiki Hayashi, University of Tsukuba, Japan |
| Genetic reagent (*D. melanogaster*) | R44E10-GAL4 | *Deady and Sun, 2015* | FBal0252601 PMID:26473732 | A gift from Jianjun Sun, University of Connecticut, USA |
| Genetic reagent (*D. melanogaster*) | RS-GAL4 | *Lee et al., 2009* | FBal0263794 PMID:19262750 | A gift from Kyung-An Han, Pennsylvania State University, USA |
| Genetic reagent (*D. melanogaster*) | nSyb-GAL4 | Bloomington Drosophila Stock Center | BDSC: #51941 RRID:BDSC_51941 FBti0154973 | FACS (5 ul per test) |
| Genetic reagent (*D. melanogaster*) | nSyb-GAL80 | *Harris et al., 2015* | PMID:26193122 | A gift from James W. Truman, Janelia Research Campus, USA |
| Genetic reagent (*D. melanogaster*) | tj-GAL4 | KYOTO stock center | DGRC: #104055 RRID:DGGR_104055 FBti0034540 | |

*Continued on next page*

*Appendix 1—key resources table continued*

| Reagent type (species) or resource | Designation | Source or reference | Identifiers | Additional information |
|---|---|---|---|---|
| Genetic reagent (*D. melanogaster*) | R13C06-GAL4 | Bloomington Drosophila Stock Center | BDSC: #47860 RRID:BDSC_47860 FBal0249828 | |
| Genetic reagent (*D. melanogaster*) | 109–30 GAL4 | Bloomington Drosophila Stock Center | BDSC: #7023 RRID:BDSC_7023 FBti0027548 | |
| Genetic reagent (*D. melanogaster*) | c355-GAL4 | Bloomington Drosophila Stock Center | BDSC: #3750 RRID:BDSC_3750 FBti0002591 | |
| Genetic reagent (*D. melanogaster*) | c306-GAL4 | Bloomington Drosophila Stock Center | BDSC: #3743 RRID:BDSC_3743 FBal0048787 | |
| Genetic reagent (*D. melanogaster*) | slbo-GAL4 | Bloomington Drosophila Stock Center | BDSC: #6458 RRID:BDSC_6458 FBst0006458 | |
| Genetic reagent (*D. melanogaster*) | bab1-GAL4 | *Bolívar et al., 2006* | FBal0242654 PMID:17013875 | A gift from Satoru Kobayashi, University of Tsukuba, Japan |
| Genetic reagent (*D. melanogaster*) | nos-GAL4 | KYOTO stock center | DGRC: #107748 RRID:DGGR_107748 FBst0306396 | |
| Genetic reagent (*D. melanogaster*) | tub > FRT > GAL80>FRT | Bloomington Drosophila Stock Center | BDSC: #38879 RRID:BDSC_38879 FBti0147580 | |
| Genetic reagent (*D. melanogaster*) | Oamb$^{KI-RD}$-GAL4 | *Deng et al., 2019* Bloomington Drosophila Stock Center | BDSC: #84677 RRID:BDSC_84677 FBti0209942 | |
| Genetic reagent (*D. melanogaster*) | Oamb-KI-T2A-GAL4 | *Kondo et al., 2020* | PMID:31914394 | |
| Genetic reagent (*D. melanogaster*) | nAChRα1-T2A-GAL4 | *Kondo et al., 2020* | PMID:31914394 | |
| Genetic reagent (*D. melanogaster*) | nAChRα2-T2A-GAL4 | *Kondo et al., 2020* | PMID:31914394 | |
| Genetic reagent (*D. melanogaster*) | nAChRα3-T2A-GAL4 | *Kondo et al., 2020* | PMID:31914394 | |
| Genetic reagent (*D. melanogaster*) | nAChRβ1-T2A-GAL4 | *Kondo et al., 2020* | PMID:31914394 | |

*Continued on next page*

*Appendix 1—key resources table continued*

| Reagent type (species) or resource | Designation | Source or reference | Identifiers | Additional information |
|---|---|---|---|---|
| Genetic reagent (*D. melanogaster*) | nAChRβ2-T2A-GAL4 | *Kondo et al., 2020* | PMID:31914394 | |
| Genetic reagent (*D. melanogaster*) | ChaT-GAL4 | Bloomington Drosophila Stock Center | BDSC: #6793 RRID:BDSC_6793 FBst0006793 | |
| Genetic reagent (*D. melanogaster*) | ppk-GAL4 | *Grueber et al., 2007* | FBtp0039691 PMID:17164414 | A gift from Hiroko Sano, Kurume University, Japan |
| Genetic reagent (*D. melanogaster*) | SPSNs-LexA | *Feng et al., 2014* | FBtp0110869 PMID:24991958 | A gift from Young-Joon Kim, Gwangju Institute of Science and Technology, South Korea |
| Genetic reagent (*D. melanogaster*) | 20xUAS-6xGFP | Bloomington Drosophila Stock Center | BDSC: #52261 RRID:BDSC_52261 FBst0052261 | |
| Genetic reagent (*D. melanogaster*) | UAS-GFP;UAS-mCD8::GFP | *Ito et al., 1997*; *Lee and Luo, 1999* | FBtp0002652 PMID:9043058 PMID:10457015 | A gift from Kei Ito, University of Cologne, Germany |
| Genetic reagent (*D. melanogaster*) | UAS-Stinger | Bloomington Drosophila Stock Center | BDSC: #84277 RRID:BDSC_84277 | |
| Genetic reagent (*D. melanogaster*) | UAS-mCD8::RFP | Bloomington Drosophila Stock Center | BDSC: #32219 RRID:BDSC_32219 FBti0131967 | |
| Genetic reagent (*D. melanogaster*) | UAS-CsChrimson | Bloomington Drosophila Stock Center | BDSC: #55134 RRID:BDSC_55134 FBti0160571 | |
| Genetic reagent (*D. melanogaster*) | UAS-Insp3R | Bloomington Drosophila Stock Center | BDSC: #30742 RRID:BDSC_30742 FBti0129829 | |
| Genetic reagent (*D. melanogaster*) | UAS-Oamb$^{K3}$ | *Lee et al., 2009* | FBtp0069415 PMID:19262750 | A gift from Kyung-An Han, Pennsylvania State University, USA |
| Genetic reagent (*D. melanogaster*) | UAS-Timp | Bloomington Drosophila Stock Center | BDSC: #58708 RRID:BDSC_58708 FBti0164930 | A gift from Andrea Page-McCaw, Vanderbilt University, USA |
| Genetic reagent (*D. melanogaster*) | UAS-nAChRα1 | This paper | | Detail described in Material and method |

*Continued on next page*

*Appendix 1—key resources table continued*

| Reagent type (species) or resource | Designation | Source or reference | Identifiers | Additional information |
|---|---|---|---|---|
| Genetic reagent (*D. melanogaster*) | *UAS > stop > dTrpA1$^{mcherry}$* | **von Philipsborn et al., 2011** | FBtp0064577 PMID:21315261 | A gift from Daisuke Yamamoto, Advanced ICT Research Institute, National Institute of Information and Communications Technology, Japan |
| Genetic reagent (*D. melanogaster*) | *UAS > stop > TNT* | **von Philipsborn et al., 2011** | FBtp0020863 PMID:21315261 | A gift from Daisuke Yamamoto, Advanced ICT Research Institute, National Institute of Information and Communications Technology, Japan |
| Genetic reagent (*D. melanogaster*) | *UAS > stop > TNT$^{in}$* | **von Philipsborn et al., 2011** | FBtp0020863 PMID:21315261 | A gift from Daisuke Yamamoto, Advanced ICT Research Institute, National Institute of Information and Communications Technology, Japan |
| Genetic reagent (*D. melanogaster*) | *dsx-FLP* | **Rezával et al., 2014** | FBal0296301 PMID:24631243 | A gift from Daisuke Yamamoto, Advanced ICT Research Institute, National Institute of Information and Communications Technology, Japan |
| Genetic reagent (*D. melanogaster*) | *TRiC; UAS-mCD8::RFP, LexAop2-mCD8::GFP;nSyb-MKII::nlsLexADBDo;UAS-p65AD::CaM* | Bloomington Drosophila Stock Center | BDSC: #61679 RRID:BDSC_61679 FBst0061679 | |
| Genetic reagent (*D. melanogaster*) | *ppk-eGFP* | **Grueber et al., 2003** | FBtp0041053 PMID:12699617 | A gift from Tadashi Uemura, Kyoto University, Japan |
| Genetic reagent (*D. melanogaster*) | *LexAop-Kir2.1* | **Feng et al., 2014** | FBtp0110870 PMID:24991958 | A gift from Young-Joon Kim, Gwangju Institute of Science and Technology, South Korea |
| Genetic reagent (*D. melanogaster*) | *UAS-LacZ$^{RNAi}$* | **Kennerdell and Carthew, 2000** | FBtp0016505 PMID:10932163 | A gift from Masayuki Miura, The University of Tokyo, Japan |
| Genetic reagent (*D. melanogaster*) | *UAS-Oamb$^{RNAi1}$* | Bloomington Drosophila Stock Center | BDSC: #31171 RRID:BDSC_31171 FBst0031171 | |

*Continued on next page*

*Appendix 1—key resources table continued*

| Reagent type (species) or resource | Designation | Source or reference | Identifiers | Additional information |
|---|---|---|---|---|
| Genetic reagent (*D. melanogaster*) | UAS-Oamb$^{RNAi2}$ | Bloomington Drosophila Stock Center | BDSC: #31233 RRID:BDSC_31233 FBst0031233 | |
| Genetic reagent (*D. melanogaster*) | UAS-Oamb$^{RNAi3}$ | Vienna Drosophila Resource Center | VDRC: #106511 FBst0478335 | |
| Genetic reagent (*D. melanogaster*) | UAS-Octβ1R$^{RNAi}$ | Vienna Drosophila Resource Center | VDRC: #110537 FBst0482104 | |
| Genetic reagent (*D. melanogaster*) | UAS-Octβ2R$^{RNAi}$ | Vienna Drosophila Resource Center | VDRC: #104524 FBst0476382 | |
| Genetic reagent (*D. melanogaster*) | UAS-Octβ3R$^{RNAi}$ | Vienna Drosophila Resource Center | VDRC: #101189 FBst0473062 | |
| Genetic reagent (*D. melanogaster*) | UAS-Insp3R$^{RNAi}$ | Bloomington Drosophila Stock Center | BDSC: #25937 FBst0025937 | |
| Genetic reagent (*D. melanogaster*) | UAS-EcR$^{RNAi}$ | Vienna Drosophila Resource Center | VDRC: #37059 FBst0461818 | |
| Genetic reagent (*D. melanogaster*) | UAS-Mmp2$^{RNAi1}$ | Bloomington Drosophila Stock Center | BDSC: #31371 RRID:BDSC_31371 FBst0031371 | |
| Genetic reagent (*D. melanogaster*) | UAS-Mmp2$^{RNAi2}$ | Vienna Drosophila Resource Center | VDRC: #330203 FBst0490996 | |
| Genetic reagent (*D. melanogaster*) | UAS-Timp$^{RNAi1}$ | Bloomington Drosophila Stock Center | BDSC: #61294 RRID:BDSC_61294 FBst0061294 | |
| Genetic reagent (*D. melanogaster*) | UAS-Timp$^{RNAi2}$ | Vienna Drosophila Resource Center | VDRC: #109427 FBst0481116 | |
| Genetic reagent (*D. melanogaster*) | UAS-Tdc2$^{RNAi1}$ | Vienna Drosophila Resource Center | VDRC: #330541 FBst0492256 | |
| Genetic reagent (*D. melanogaster*) | UAS-Tdc2$^{RNAi2}$ | Bloomington Drosophila Stock Center | BDSC: #25871 RRID:BDSC_25871 FBst0025871 | |
| Genetic reagent (*D. melanogaster*) | UAS-Tβh$^{RNAi1}$ | Vienna Drosophila Resource Center | VDRC: #107070 FBst0478893 | |

*Continued on next page*

*Appendix 1—key resources table continued*

| Reagent type (species) or resource | Designation | Source or reference | Identifiers | Additional information |
|---|---|---|---|---|
| Genetic reagent (*D. melanogaster*) | UAS-T*βh*RNAi2 | Bloomington Drosophila Stock Center | BDSC: #67968 RRID:BDSC_67968 FBst0067968 | |
| Genetic reagent (*D. melanogaster*) | UAS-ChATRNAi1 | Vienna Drosophila Resource Center | VDRC: #330291 FBst0490951 | |
| Genetic reagent (*D. melanogaster*) | UAS-ChATRNAi2 | Bloomington Drosophila Stock Center | BDSC: #25856 RRID:BDSC_25856 FBst0025856 | |
| Genetic reagent (*D. melanogaster*) | UAS-nAChR*α*1RNAi | Vienna Drosophila Resource Center | VDRC #48159 FBst0467755 | |
| Genetic reagent (*D. melanogaster*) | UAS-nAChR*α*2RNAi | Vienna Drosophila Resource Center | VDRC: #101760 FBst0473633 | |
| Genetic reagent (*D. melanogaster*) | UAS-nAChR*α*3RNAi | Vienna Drosophila Resource Center | VDRC: #101806 | |
| Genetic reagent (*D. melanogaster*) | UAS-nAChR*β*1RNAi | Vienna Drosophila Resource Center | VDRC: #106570 FBst0478394 | |
| Genetic reagent (*D. melanogaster*) | UAS-nAChR*β*2RNAi | Vienna Drosophila Resource Center | VDRC: #109450 FBst0481138 | |
| Genetic reagent (*D. melanogaster*) | UAS-nvdRNAi1 | *Yoshiyama et al., 2006* | FBal0193613 PMID:16763204 | |
| Genetic reagent (*D. melanogaster*) | UAS-nvdRNAi2 | *Yoshiyama et al., 2006* | FBal0193614 PMID:16763204 | |
| Chemical, compound, drug | Octopamine | Sigma-Aldrich | #O0250 | |
| Chemical, compound, drug | Schneider's *Drosophila* medium | Thermo Fisher Scientific | #21720024 | |
| Chemical, compound, drug | 20-hydroxyecdysone | Enzo Life Sciences | ALX-370–012 | |
| Antibody | anti-GFP (chicken polyclonal) | Abcam | #ab13970 | 1:4000 dilution |
| Antibody | anti-RFP (rabbit polyclonal) | Medical and Biological Laboratories | #PM005 | 1:2000 dilution |

*Continued on next page*

*Appendix 1—key resources table continued*

| Reagent type (species) or resource | Designation | Source or reference | Identifiers | Additional information |
|---|---|---|---|---|
| Antibody | anti-Hts 1B1 (mouse monoclonal) | Developmental Studies Hybridoma Bank | | 1:50 dilution |
| Antibody | anti-DE-cadherin DCAD2 (rat monoclonal) | Developmental Studies Hybridoma Bank | | 1:50 dilution |
| Antibody | anti-pH3 (rabbit polyclonal) | Merck Millipore | #06–570 | 1:2000 dilution |
| Antibody | anti-pMad (rabbit polyclonal) | Abcam | #ab52903 | 1:2000 dilution |
| Antibody | anti-Lamin C LC28.26 (mouse monoclonal) | Developmental Studies Hybridoma Bank | | 1:10 dilution |
| Antibody | anti-cleaved Dcp-1 (rabbit polyclonal) | Cell Signaling Technology | #9578 | 1:1000 dilution |
| Antibody | anti-Vasa (rat monoclonal) | Developmental Studies Hybridoma Bank | | 1:50 dilution |
| Antibody | anti-LacZ 40-1a (mouse monoclonal) | Developmental Studies Hybridoma Bank | | 1:50 dilution |
| Antibody | anti-Tdc2 (rabbit polyclonal) | Abcam | #ab128225 | 1:2000 dilution |
| Antibody | Alexa Fluor 546 phalloidin | Thermo Fisher Scientific | #A22283 | 1:200 dilution |
| Antibody | Alexa Fluor 633 phalloidin | Thermo Fisher Scientific | #A22284 | 1:200 dilution |
| Chemical, compound, drug | FluorSave reagent | Merck Millipore | #345789 | |
| Chemical, compound, drug | all trans-Retinal | Sigma-Aldrich | #R2500 | |
| Software, algorithm | ImageJ | | https://imagej.nih.gov/ij; RRID:SCR_003070 PMID:22930834 | |
| Software, algorithm | R | | RRID:SCR_001905 | |

