## [Decision Letter]

**Acceptance summary:**

In this manuscript, Niwa and colleagues investigated the mechanism of octopaminergic system in regulating mating-induced GSC increase, a phenomenon previously identified by the same group. A fundamental question remaining in the field is how SPR-positive sensory neurons (SPSNs) relays the mating signal to GSC to increase their numbers after mating. These authors found that *Oamb*, one of the octopamine (OA) receptors, is expressed in escort cells in the germarium. Using sophisticated genetic tools, they illustrated that OA/Oamb signaling in escort cells is required and sufficient for mating induced GSC increase. OA/Oamb-induced GSC increase depends on Ca^2+^ increase, ecdysone signaling, and *Mmp2*, similar to the OA-induced follicle rupture in mature follicles. They went on to identify a subset of OA neurons (*dsx*^+^*Tdc2*^+^) in the abdominal ganglion responsible for OA signaling in escort cells and GSC increase. Furthermore, they illustrated that SPSNs directly innervate *dsx*^+^*Tdc2*^+^ neurons and that acetylcholine released from SPSNs inhibits OA release from *dsx*^+^*Tdc2*^+^ neurons, which could be overridden by mating. Therefore, they established a clear relay mechanism from mating to SPSNs, *dsx*^+^*Tdc2*^+^ neurons, *Oamb* signaling in escort cells, and GSC increase.

Overall, all experiments were well designed and included all the controls. The data were interpreted accurately and support their conclusions in general. The work is a nice addition to our understanding of extrinsic cues to stem cell regulation.

**Decision letter after peer review:**

Thank you for submitting your article "Neuronal octopamine signaling regulates mating-induced germline stem cell proliferation in female *Drosophila*" for consideration by *eLife*. Your article has been reviewed by three peer reviewers, including Yukiko M Yamashita as the Reviewing Editor and Reviewer #1, and the evaluation has been overseen by Utpal Banerjee as the Senior Editor. The following individual involved in review of your submission has agreed to reveal their identity: Jianjun Sun (Reviewer #2).

The reviewers have discussed the reviews with one another and the Reviewing Editor has drafted this decision to help you prepare a revised submission.

Summary:

In this manuscript, Niwa and colleagues investigated the mechanism of octopaminergic system in regulating mating-induced GSC increase, a phenomenon previously identified by the same group. A fundamental question remaining in the field is how SPR-positive sensory neurons (SPSNs) relays the mating signal to GSC to increase their numbers after mating. These authors found that *Oamb*, one of the octopamine (OA) receptors, is expressed in escort cells in the germarium. Using sophisticated genetic tools, they illustrated that OA/Oamb signaling in escort cells is required and sufficient for mating induced GSC increase. OA/Oamb-induced GSC increase depends on Ca^2+^ increase, ecdysone signaling, and *Mmp2*, similar to the OA-induced follicle rupture in mature follicles. They went on to identify a subset of OA neurons (*dsx*^+^*Tdc2*^+^) in the abdominal ganglion responsible for OA signaling in escort cells and GSC increase. Furthermore, they illustrated that SPSNs directly innervate *dsx*^+^*Tdc2*^+^ neurons and that acetylcholine released from SPSNs inhibits OA release from *dsx*^+^*Tdc2*^+^ neurons, which could be overridden by mating. Therefore, they established a clear relay mechanism from mating to SPSNs, *dsx*^+^*Tdc2*^+^ neurons, *Oamb* signaling in escort cells, and GSC increase.

Overall, all experiments were well designed and included all the controls. The data were interpreted accurately and support their conclusions in general. The work is a nice addition to our understanding of extrinsic cues to stem cell regulation. The reviewers identified the following major concerns, which should be addressed in the revision. Given the pandemic, the reviewers feel it to be appropriate if the statements are weakened, acknowledging the caveats in interpretation.

Essential revisions (which can be done with textual changes, unless they do not already have the data):

1) The evidence that the relevant target cells of the octopaminergic neurons are escort cells is rather weak; The authors cannot rule out contributions from follicle cells based on the current data. The authors use drivers expressed in both cell types (*C587-Gal4*, *Tj-gal4*) for most knockdown/overexpression experiments. It will be important to exclude this possibility by using a *Gal4* driver only expressed in early follicle cells. (*R44E10-Gal4* used to exclude other cell types is expressed in follicle cells of stage 14 oocytes---which is too late to assess the involvement of follicle cells). Alternatively, the authors should weaken the statement to be precise on this point.

2) Inferred from *Timp* experiments, authors conclude that *Mmp2* activity in the GSC niche cells is necessary for mating-induced GSC increase (subsection “Matrix metalloproteinase 2 acts downstream of OA-Oamb-Ca^2+^ signaling”, second paragraph). It will be important to examine the *Mmp2* expression and activity in both virgin and mated female to see whether *Mmp2* activity is indeed regulated by mating. The authors seem to alternate between two models. Throughout most of the manuscript, the mating signal received by the ovary seems to be the sex peptide-regulated octopamine release. However, in the section where the *Mmp2*/*Timp1* data is presented, the implication is that *Timp1* expression would go down after mating, resulting in increased *Mmp2* activity. Do *Timp1* or any other members of the OA-to-*Mmp2* signalling axis change after mating? How would the authors reconcile/integrate the two models? Related to this, the epistasis between ecdysone signalling and *Mmp2* activity is unclear.

---

## [Author Response]

Essential revisions (which can be done with textual changes, unless they do not already have the data):1) The evidence that the relevant target cells of the octopaminergic neurons are escort cells is rather weak; The authors cannot rule out contributions from follicle cells based on the current data. The authors use drivers expressed in both cell types (C587-Gal4, Tj-gal4) for most knockdown/overexpression experiments. It will be important to exclude this possibility by using a Gal4 driver only expressed in early follicle cells. (R44E10-Gal4 used to exclude other cell types is expressed in follicle cells of stage 14 oocytes---which is too late to assess the involvement of follicle cells). Alternatively, the authors should weaken the statement to be precise on this point.

According to the reviewers’ suggestion, we did additional RNAi experiments using *c355-GAL4, c306-GAL4, slbo-GAL4, R13C06-GAL4*, and *109-30-GAL4*. Each of these strains expresses *GAL4* in distinct cell types of ovarian somatic cells. We found that *Oamb* RNAi in the stage-14 follicle cells by *R44E10-GAL4*, or in the stage 9-10 follicle cells by *c355-GAL4*, *c306-GAL4*, and *slbo-GAL4*, had no significant effect on mating-induced GSC increase. These results suggest that *Oamb* in the mature follicle cells is not involved in mating-induced GSC increase. Therefore, it is quite unlikely that *Oamb* in the follicle cells after stage 9 has a large contribution to regulating mating-induced GSC increase.

On the other hand, we found that *Oamb* RNAi by *R13C06-GAL4,* and *109-30-GAL4*, which are active in the escort cells, and the germarium follicle cells, respectively, resulted in the failure of mating-induced GSC increase. Along with the data obtained from *Tj-GAL4* and *c587-GAL4* drivers, we conclude that *Oamb* in the in the escort cells or the follicle cells of the germarium, plays an essential role in mating-induced GSC increase.

These additional data are described in the subsection “Mating-induced GSC increase requires the octopamine receptor *Oamb* in ovarian escort cells” and represented in Figure 1—figure supplement 1C, D and E of the revised manuscript.

2) Inferred from Timp experiments, authors conclude that Mmp2 activity in the GSC niche cells is necessary for mating-induced GSC increase (subsection “Matrix metalloproteinase 2 acts downstream of OA-Oamb-Ca^2+^ signaling”, second paragraph). It will be important to examine the Mmp2 expression and activity in both virgin and mated female to see whether Mmp2 activity is indeed regulated by mating. The authors seem to alternate between two models. Throughout most of the manuscript, the mating signal received by the ovary seems to be the sex peptide-regulated octopamine release. However, in the section where the Mmp2/Timp1 data is presented, the implication is that Timp1 expression would go down after mating, resulting in increased Mmp2 activity. Do Timp1 or any other members of the OA-to-Mmp2 signalling axis change after mating? How would the authors reconcile/integrate the two models? Related to this, the epistasis between ecdysone signalling and Mmp2 activity is unclear.

We appreciate the reviewer’s comments and wish to allay any reservations. First, we did not succeed in obtaining any solid data to support the idea that *Timp* expression is changed before and after mating. Nevertheless, we admittedly overstated the role of *Timp* in mating-induced GSC increase in our original manuscript. Therefore, we have deleted this sentence from the revised manuscript. We apologize for any confusion or overstatements.

Second, regarding *Mmp2* activity, we tried to examine whether *Mmp2* protein levels were changed before and after mating with *Mmp2*-EGFP strain (Deady et al., 2015) and also with anti-*Mmp2* antibody (Dear et al. Development 2016). However, unfortunately, we could not detect any obvious signals in the germarium region of either virgin or mated females in our hands. It must be noted, however, that previous studies detected *Mmp2* expression in the germarium (Pearson et al., 2016 and Wang and Page-McCaw, 2014). Currently we are unsure as to the reason why we could not detect these signals in our experimental conditions, but the COVID-19 situation prevents us from promptly solving this problem during this revision process. Therefore, we did not include our negative data in the revised manuscript, but just cited the previous studies (Pearson et al., 2016 and Wang and Page-McCaw, 2014) in the subsection “Matrix metalloproteinase 2 acts downstream of OA-Oamb-Ca^2+^ signaling”.

Regarding another question about the epistasis between ecdysone signaling and *Mmp2* activity, we did not have a clear answer but faithfully discussed this point in the subsection “The role of Mmp2 in mating-induced GSC increase”.

We would like to emphasize again that *Timp* overexpression significantly suppressed mating-induced GSC increase in our model system (Figure 4B), strongly suggesting that *Mmp2* is involved in this process.